# LLM-based Typed Hyperresolution for Commonsense Reasoning with Knowledge Bases

**Armin Toroghi**[1], **Ali Pesaranghader**[2], **Tanmana Sadhu**[2], **Scott Sanner**[1,3]
[1]University of Toronto, [2]LG Electronics, Toronto AI Lab, [3]Vector Institute of AI
`armin.toroghi@mail.utoronto.ca`,
`{ali.pesaranghader, tanmana.sadhu}@lge.com` ,`ssanner@mie.utoronto.ca`

## Abstract

Large language models (LLM) are being increasingly applied to tasks requiring commonsense reasoning. Despite their outstanding potential, the reasoning process of LLMs is prone to errors and hallucinations that hinder their applicability, especially in high-stakes scenarios. Several works have attempted to enhance commonsense reasoning performance of LLMs by (i) using prompting styles that elicit more accurate reasoning, (ii) utilizing the LLM as a semantic parser for a symbolic reasoner, or (iii) enforcing the LLM to simulate a logical inference rule. However, all these solutions have critical limitations: they are unable to leverage the internal commonsense knowledge of the LLM in tandem with an axiomatic knowledge base, they lack a mechanism to reliably repair erroneous inference steps, and their application is restricted to small knowledge bases that fit the context limit of the LLM. In this work, we present LLM-based Typed Hyperresolution (LLM-TH), a logical commonsense reasoning framework that leverages *"theory resolution"*, a concept from classical logical inference which enables integrating LLMs into the *"resolution"* inference rule, thus mitigating reasoning errors and hallucinations and enabling verification of the reasoning procedure. LLM-TH is also equipped with a mechanism for repairing erroneous inference steps supported by theoretical guarantees. Using *"Hyperresolution"* and *"Typed inference"* schemes, we show that LLM-TH can efficiently reason over large knowledge bases consisting of tens of thousands of rules with arbitrary predicate arities. Our experiments on three diverse language-based reasoning tasks—preference reasoning, multi-domain deductive reasoning, and geographical question answering—showcase that LLM-TH, using merely a BART 406M parameter NLI entailment model, significantly reduces reasoning errors compared to baselines using Llama3-70B, Gemini1.5-Flash, GPT-3.5-Turbo, and Mixtral-46.7B.

## 1 Introduction

The breakthrough in Large Language Models (LLMs) has significantly impacted AI research, paving the way for deploying AI-powered systems in various tasks and applications. This huge impact is primarily due to the outstanding performance of LLMs in tasks that require substantial reasoning skills (Chang et al., 2024; Plaat et al., 2024). LLMs have also acquired commonsense understanding, a critical element for interacting with the real world (Zhao et al., 2024; Valmeekam et al., 2024). However, reasoning performance of LLMs is not infallible. LLMs commonly show reasoning errors and make hallucinations—generating incorrect outputs that seem valid—which hinders their reliable deployment, particularly in high-risk tasks (Tonmoy et al., 2024; Zhang et al., 2023b).

To overcome these challenges in LLM-based reasoning, several approaches have been proposed in the literature that can be broadly categorized into three groups: (i) Using prompting styles that can elicit more accurate reasoning from the LLM (Wei et al., 2022; Kojima et al., 2022; Zhou et al., 2022) or augmenting the prompt by retrieved information (Lewis et al., 2020b), (ii) using the LLM to translate natural language problem and knowledge bases (KB) for a symbolic logical solver (Olausson et al., 2023; Pan et al., 2023), and (iii) using the LLM to emulate a logical inference rule to solve the reasoning problem (Kazemi et al., 2023; Lee & Hwang, 2024).

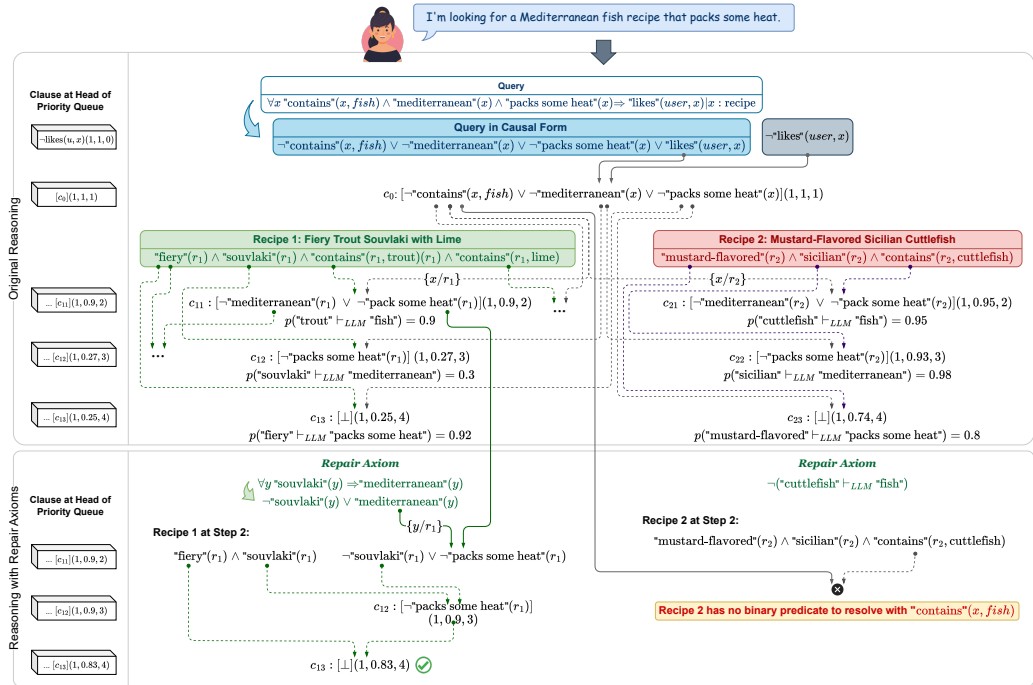

Figure 1: Workflow of LLM-TH shown with a preference reasoning example. Top: Using LLM-based typed hyperresolution to compute proof scores for each recipe option to entail user query. Negated query is the first active clause, and each resolvent is assigned a priority tuple: (*type entailment score*, *predicate entailment score*, *proof length score*) and pushed to the priority queue (only the foremost clause is shown for priority queue of recipe 1 which becomes the next active clause). LLM makes two mistakes: assigning a low score to *"Souvlaki"* entailing *"Mediterranean"* and a high score to *"cuttlefish"* entailing *"fish"*. Bottom: Both kinds of mistakes can be fixed after the insertion of repair axioms, resulting in the correct recommendation of Recipe 1.

These works have advanced the logical reasoning performance of LLMs; yet, they are hindered by a number of important limitations: (a) Their application is limited to small KBs that can fit into the context limit of the LLM, and are not scalable to practical KBs such as the widely-used Knowledge Graphs (KG) containing thousands of facts and axioms. (b) They are restricted to reason on a complete KB containing all rules required to solve the problem. However, assuming access to such KB is typically unrealistic in practical use cases, thus calling for a methodology to leverage the internal commonsense knowledge of the LLM in the reasoning process. (c) All steps involved in the reasoning process are not transparent and thus, the correctness of the final answer cannot be determined by inspecting the reasoning process. (d) Upon observation of a reasoning error, they do not provide any reliable framework to fix the error and ensure it will not occur in the future.

In this work, we aim to address these limitations by making the following contributions:

- We introduce LLM-based Typed Hyperersolution (LLM-TH), a framework for efficient complex logical commonsense reasoning with KBs containing predicates of arbitrary arities, that facilitates the incorporation of the internal commonsense knowledge of LLMs in the reasoning process. LLM-TH is founded on *"theory resolution"* (Stickel, 1985; Baumgartner, 1992), a concept from classical logical reasoning that allows for the incorporation of specialized theorem provers into the resolution inference rule. (Section 3.1)

- We equip LLM-TH with a mechanism for incorporating the type information of the variables and constants in the problem domain to prune the proof search space and terminate the exploration of reasoning paths that are unlikely to succeed at very early stages. Also, using hyperresolution, an extension of resolution that enables combining clauses to perform several resolution steps simultaneously, we make LLM-TH an efficient and scalable

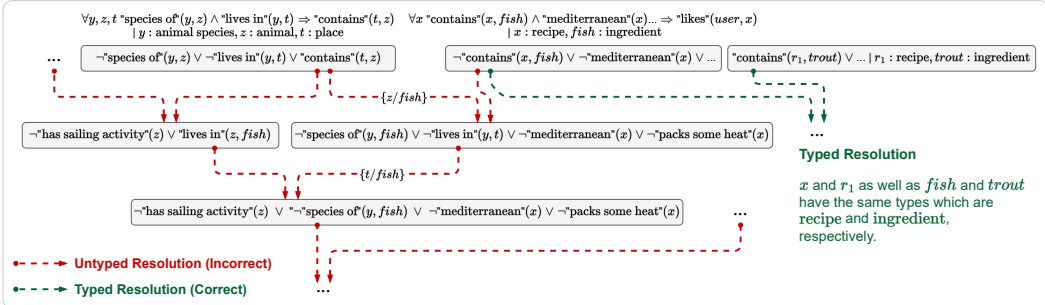

Figure 2: An example of typing mechanism of LLM-TH. The binary predicate *"contains"* is common among clauses from different domains. Left: Application of resolution rule without considering types leads to resolving a literal about *"recipes"* with complementary literals from domains such as *"animals"*. Each incorrect resolution results in a new branch that will be explored but leads to failure, making the process inefficient. Right: in typed resolution, only literals with consistent variable and constant types will be unified, therefore pruning the search space and enhancing efficiency.

reasoning framework for logical commonsense reasoning with LLMs. We show that LLM-TH is easily scalable to KBs consisting of tens of thousands of rules. (Section 5.2.1)

- We show that by providing access to the exact axioms and facts used at every reasoning step, LLM-TH results in a verifiable and faithful reasoning performance. Furthermore, we propose a reliable repair methodology for missed inferences and incorrect reasoning steps due to LLM hallucinations and missed inferences, and provide theoretical proof that it reliably fixes reasoning errors. (Section 4)

- We show that by using the theory hyperresolution framework, LLM-TH is able to leverage the internal commonsense knowledge of the LLM to compensate for KB incompleteness and perform accurate and reliable reasoning. (Section 5.2.2)

- We experiment with LLM-TH on three different tasks involving commonsense reasoning: preference reasoning, multi-domain deductive reasoning, and geographical QA, showcasing the superiority of LLM-TH in terms of answer and reasoning accuracy over Chain of Thought (Kojima et al., 2022; Wei et al., 2022) and retrieval augmented generation (RAG) (Lewis et al., 2020b) baselines using orders of magnitude larger LLMs. (Section 5)

## 2 RELATED WORKS

**Enhancing LLM Reasoning** As LLMs scale, they exhibit emergent behaviors such as the capability of solving problems that involve reasoning (Chang et al., 2024; Huang & Chang, 2022). However, their reasoning performance often suffers from errors and hallucination of facts in their inferences (Tonmoy et al., 2024; Zhang et al., 2023b). Several works have shown that with certain prompting techniques such as Chain of Thought (CoT) (Wei et al., 2022; Kojima et al., 2022), Tree of thought (Yao et al., 2024), Selection-Inference (Creswell et al., 2022), Self-consistency (Wang et al., 2022), Least-to-most prompting (Zhou et al., 2022), etc., more accurate reasoning can be elicited from LLMs. Retrieval Augmented Generation (RAG) (Lewis et al., 2020b) has also been noted as an effective approach in reducing reasoning errors and hallucinations by including relevant content retrieved from a KB in the prompt to condition LLM's reasoning on dependable information. These methods have made significant progress in advancing the reasoning behavior of LLMs, but even applying them does not guarantee an accurate reasoning behavior from the LLM. Furthermore, since the LLM is entirely in charge of doing the reasoning, there is no control over the reasoning process and its correctness cannot be verified (Shanahan, 2024; Pan et al., 2023). Moreover, the performance of these methods has been shown to degrade when being applied to long-horizon (Dziri et al., 2024) and out-of-domain (Saparov et al., 2024) reasoning tasks as well as problems involving negation (Anil et al., 2022) and contraposition (Zhang et al., 2024).

**Formal Reasoning with LLMs** To offer more control over the reasoning process, two groups of work have been proposed for formal reasoning with LLMs: (i) *Semantic parsing* methods remove

the LLM from reasoning and only use it to translate the problem into a symbolic format and delegate the reasoning task to a symbolic solver (Pan et al., 2023; Olausson et al., 2023). (ii) Works enforcing the LLM to emulate an established logical inference rule such as backchaining (Kazemi et al., 2023; Lee & Hwang, 2024). While these works make progress in mitigating hallucinations, they rely on the existing KB rules and have no mechanism to use the rich commonsense knowledge of the LLM in their reasoning. Although recent works (Toroghi et al., 2024a) used resolution inference rule for LLM-based reasoning, they are restricted to unary predicates and small KBs.

**Commonsense Reasoning with LLMs**   We refer the reader to Appendix F.

## 3   METHODOLOGY

We first review the (hyper)resolution rule for inference in first-order logic using a function and equality-free first-order logical (FOL) syntax in clausal normal form (Chang & Lee, 2014). We then proceed to introduce the concept of Theory resolution that leverages external sources of knowledge (such as LLMs) to augment the resolution process. Next, we introduce the concept of typed resolution that will be critical for improving the efficiency of inference in our work by prioritizing inference with compatible types and reducing proof length. With all of these ingredients in place, we conclude with our LLM Theory Hyperresolution algorithm (LLM-TH) along with crucial propositions that ensure the correctness of *repair axioms* used to override incorrect LLM inferences.

**Resolution Rule and Hyperresolution**   *Resolution* is a sound and complete inference rule which is widely used in logical reasoning. From two premise clauses containing complementary literals, resolution rule derives a *resolvent* clause by canceling (resolving) the complementary literals, e.g.,

$$\frac{A(x) \vee B(x,y) \quad \neg B(w,z) \vee C(z)}{A(x) \vee C(y)} \ , \tag{1}$$

under the unification $\theta = \{x/w, y/z\}$. Repeated application of the resolution rule will either result in a contradiction, e.g., deriving both $A(x)$ and $\neg A(x)$ indicating an inconsistent clause set, or reaching a point where no further resolutions are possible.

The efficiency of the repeated application of the resolution rule can be substantially improved by *hyeperresolution* (Robinson, 1965), an extension of resolution that enables combining multiple resolution steps in one inference step. Concretely, it resolves positive literals with all possible matching negative clauses simultaneously, e.g.,

$$\frac{A(x) \vee B_1(x,y)... \vee B_n(x,y) \quad C(z) \vee \neg B_1(w,z) \vee ... \vee \neg B_n(w,z)}{A(x) \vee C(y)} \ , \theta = \{x/w, y/z\}. \tag{2}$$

**LLM-based Theory Resolution**   Application of the resolution rule is originally restricted to clauses with complementary literals that share identical predicates. Theory resolution (Stickel, 1985; Baumgartner, 1992) relaxes this condition and broadens the applicability of the resolution rule by integrating special-purpose theories into resolution. Based on theory resolution, given two clauses $c_1 = A(x) \vee B(x,y)$ and $c_2 = \neg C(w,z) \vee D(z)$, if a theorem prover $T$ identifies $B(x,y)$ and $\neg C(w,z)$ under unification $\theta = \{x/w, y/z\}$ to be unsatisfiable (i.e., $\forall x \forall y B(x,y) \wedge \neg C(x,y) \vdash_T \perp$), the clauses can be resolved despite lacking complimentary literals with identical predicates:

$$\frac{A(x) \vee B(x,y) \quad \neg C(w,z) \vee D(z)}{A(x) \vee D(y)} \ , \theta = \{x/w, y/z\}. \tag{3}$$

In this work, we use an LLM as the theory that identifies the unsatisfiable natural language predicates to perform reasoning via theory resolution. Translating natural language to symbolic form, as semantic parsing methods do, is substantially restricted. For example, semantic parsing will map *"packs some heat"* and being *"spicy"* to completely different symbolic predicates. Therefore, a symbolic reasoner is unable to discern their entailment relationship unless given explicit axioms.

Using LLM-based theory resolution, we can integrate the LLM's commonsense knowledge into the reasoning process to find entailments between predicates and constants without requiring explicit axioms. We do this in an extended version of FOL in which predicates, functions, and constants

are no longer symbols, but natural language text. In this logical system, which we call *natural language logic*, the unsatisfiability condition in theory resolution reduces to natural language entailment. In other words, if an LLM identifies a natural language predicate $B$ to entail predicate $D$, i.e., $B(x) \vdash_{LLM} D(x)$, and therefore, $B(x) \wedge \neg D(x) \vdash_{LLM} \bot$, then literals $B(x)$ and $D(x)$ can be resolved. For instance, given clauses $c_1 =$ *"packs some heat"*$(x)$ and $c_2 = \neg$*"spicy"*$(x) \vee Q(x)$, in which $Q(x)$ is another literal with a natural language predicate, since the LLM identifies the natural language entailment *"packs some heat"* $\vdash_{LLM}$ *"spicy"*, a theory resolution step can be performed as

$$\frac{\text{``packs some heat''}(x) \quad \neg \text{``spicy''}(x) \vee Q(x)}{Q(x)} \ . \tag{4}$$

**LLM-based Typed Theory Resolution** Although resolution is one of the most widely used inference rules in logical reasoning with the key properties of being sound and complete, its application on real-world large-scale knowledge bases is computationally expensive. One of the key challenges is that the space of possible resolutions—the combinations of literals and clauses that can potentially be unified and resolved—can become enormous, and grows exponentially during the resolution process, leading to inefficiencies in finding contradictions or valid derivations. This challenge is often addressed by choosing particular orderings that first explore resolution steps that are more likely to lead to valid proofs (Baumgartner, 1992; Sanner & McIlraith, 2006).

In addition to proposing an ordering strategy which is explained in 3.1, we introduce the notion of typing into theory resolution which considerably prunes the space of possible resolutions. Many of the created resolvents during the resolution process, each opening a new search branch, are created by unifying variables and constants with inconsistent ontological types. For instance, a literal *"small"(x)* in a clause about *vehicles* $x$ can be resolved with a literal $\neg$*"small"(y)* which is about *animals* $y$. However, the search branch created by this resolvent is very unlikely to result in a valid proof as it uses a semantically unlikely unification (*vehicle-animals*) that is likely to lead to a failed inference in subsequent steps. By associating variables and constants with their ontological types and considering type consistency in unification, we can significantly prune the space of allowable resolutions, by preventing the exploration of type inconsistent branches from the beginning.

Variable types can be introduced into an FOL sentence as unary predicates, e.g., *"animal"*$(x)$. Considering a universally quantified sentence in FOL as

$$\forall x \forall y \ H(x) \wedge T(y) \wedge A(x,y) \implies R(x,y), \tag{5}$$

where $H(.)$ and $T(.)$ are unary predicates indicating types of their corresponding variables, we equivalently write the above sentence in the *typed* FOL as

$$\forall x \forall y \ A(x,y) \implies R(x,y)|x:H, y:T, \tag{6}$$

where, $x:H$ and $y:T$ indicate that $x$ and $y$ are of type $H$ and $T$ respectively. This clause can be equivalently written in the clausal form as

$$\forall x \forall y \ \neg A(x,y) \vee R(x,y)|x:H, y:T, \tag{7}$$

Next, consider another clause in typed FOL as

$$\forall w \forall z \ \neg R(w,z) \vee S(w,z)|w:M, z:N. \tag{8}$$

Using the unifier $\theta = \{w/x, z/y\}$, we can perform *typed resolution* between the clauses 7 and 8 as

$$\frac{\neg A(x,y) \vee R(x,y)|x:H, y:T \quad \neg R(w,z) \vee S(w,z)|w:M, z:N}{\neg A(x,y) \vee S(x,y)|x:C, y:D} \ , \tag{9}$$

where $C$ and $D$, the types of resolvent variables are $C \equiv H \sqcap M$ and $D \equiv T \sqcap N$ using the following lemma which is proved in Appendix A. Here, $\sqcap$ indicates unary type intersection (conjunction).

**Lemma 1.** *Resolving two disjunctive clauses $c_1$ and $c_2$ that include complimentary literals $l(x_1,...,x_n)|x_1 : T_1,...,x_n : T_n$ and $\neg l(y_1,...,y_n)|y_1 : H_1,...,y_n : H_n$ under the unifier $\theta = \{x_1/y_1,...,x_2/y_2\}$ creates a resolvent $c_{res}$ with type variables $x_1 : T_1 \sqcap H_1,...,x_n : T_n \sqcap T_n$.*

Typed resolution can be directly extended to typed theory resolution in our natural language-based logical system by resolving literals $B(x_1,...,x_n)|x_1 : T_1,...,x_n : T_n$ and $D(y_1,...,y_n)|y_1 : H_1,...,H_n$ if the LLM identifies the natural language predicate $B$ to entail $D$, i.e., $B(x_1,...,x_n) \vdash_{LLM} D(y_1,...,y_n)$ if their unified variables have consistent types. In the next section, we elaborate on how type consistency is checked in our LLM-TH framework.

---

**Algorithm 1** LLM-TH Algorithm

---

1: **Input:** $\mathcal{K}$, $q$, *max_proofs*, *max_iters*, $\mathcal{I}$, $\mathcal{F}$
2: *proofs* $\leftarrow \emptyset$
3: $PQ \leftarrow \emptyset$ // $PQ$ is an initially empty priority queue.
4: $PQ.push(\neg q, (1, 1, 0))$ // Negation of $q$ has priority $(1, 1, 0)$, $PQ$ is ordered by Equation 15
5: $i \leftarrow 0$
6: **while** $(PQ \neq \emptyset) \wedge (|proofs| \leq max\_proofs) \wedge (i \leq max\_iters)$ **do**
7:     $c \leftarrow PQ.pop()$
8:     **if** $c = \bot$ **then**
9:         *proofs* $\leftarrow$ *proofs* $\cup \{(\mathcal{A}_c, (\rho^t(c), \rho^e(c), \rho^l(c)))\}$ // $\mathcal{A}_c$ is the set of ancestors of $c$, found by backtracking up to $\neg q$
10:     **else**
11:         $\beta_c \leftarrow$ candidate clauses in $\mathcal{K}$ with similar arity and different polarity to $c$
12:         **for** $c_{\text{target}} \in \beta_c$ **do**
13:             Perform hyperresolution to compute resolvent $c_{\text{res}}$ of $c$ and $c_{\text{target}}$ using Equation 2
14:             $PQ.push(c_{\text{res}}, (\rho^t(c_{\text{res}}), \rho^e(c_{\text{res}}), \rho^l(c_{\text{res}})))$ // *cf. Equations 11, 13 and 14*
        $i \leftarrow i + 1$
15: **Output:** *proofs*

---

## 3.1 LLM-TH ALGORITHM

We present LLM-TH, an algorithm for efficient logical commonsense reasoning that extends theory resolution to LLM-based reasoning with predicates of arbitrary arity, hyperresolution and type inference. Its workflow is presented in Figures 1 and 2, Algorithm 1, and explained in Appendix E.

**Problem Definition** Let $\mathcal{Q}$ denote a set of queries and $\mathcal{K}$ represent a knowledge base (KB) comprising a set of axioms and facts, a set of intended repair axioms $\mathcal{I}$, and a set of forbidden repair axioms $\mathcal{F}$, all expressed in natural language logic and clausal form with arbitrary predicate arities. For each query $q \in \mathcal{Q}$, LLM-TH identifies a set of proofs, denoted as *proofs*. Each proof $f \in$ *proofs* consists of a subset of clauses in $\mathcal{K}$, and is assigned a priority score $\rho$.

**Algorithm** To prove $\mathcal{K}$ entails the query q via resolution, we need to show that repeatedly using the resolution rule on $\mathcal{K} \wedge \neg q$ leads to a contradiction, and is thus unsatisfiable. Following the backward chaining paradigm that offers efficiency benefits (Poole & Mackworth, 2010), we pick $\neg q$ as the first active clause to be resolved with a clause from $\mathcal{K}$. At each step, given an active clause $c = \bigvee_{i=1}^{|c|} l_i$ where each $l_i$ is a literal of arbitrary arity, any clause $c_{target} \in \mathcal{K}$ as $c_{target} = \bigvee_{i=1}^{|c_{target}|} l_{target_i}$ is considered a *candidate clause* to be theory resolved with $c$, yielding the resolvent $c_{res}$, if at least one $(l_i, l_{target_i})$ pair can be formed where $l_i$ and $l_{target_i}$ have equal arities and different polarities. One can lift this polarity restriction for theory resolutions to allow implicit negation in the language of the predicate (e.g.,"*not spicy*") in contrast to explicit logical negation (e.g., $\neg$"*spicy*"). As $\mathcal{K}$ is often large and expands further with new resolvents being derived as resolution advances, efficiency is a key desideratum, which LLM-TH achieves by prioritizing candidate clauses based on two criteria: (i) type consistency and (ii) predicate entailment between the active and candidate clauses.

*Restricting the Space of Resolutions with Typing:* The first mechanism used by LLM-TH to improve efficiency is the use of typed theory resolution to restrict the space of allowable resolutions. In typed theory resolution, the types of variables in the resolvent clause are determined by the conjunction of the types of variables in their parent clause, hence LLM-TH prioritizes clauses with variable types that align with the active clause to encourage valid types. For example, if two candidate clauses are considered to be resolved with an active clause of variable type *"Animal"*, LLM-TH prioritizes a candidate clause with a unifying variable type *"Mammal"* over one with type *"Vehicle"*.

LLM-TH leverages the commonsense knowledge of the LLM to obtain the probability of entailment between the variable types. Formally, for the pair of literals with equal arities $(l, l_{target})$, denoting the set of argument types of $l$ and $l_{target}$ as $T = \{t_i\}$ and $T' = \{t'_i\}$ respectively, the plausibility of unifying each of their variables can be obtained by calculating the entailment probabilities between each $t_i$ and $t'_i$. Since entailment is an asymmetric relation and its direction is unknown, we need to

calculate both $t_i \vdash_{LLM} t_i'$ and $t_i' \vdash_{LLM} t_i$ to obtain the type consistency score. The average of type entailment scores for arguments determines $\rho^{type}(c_{res})$, the overall type priority score for $c_{res}$.

$$\rho^{type}(c_{res}) = \frac{1}{|T_i|} \Sigma_i \left( \max\{p(t_i \vdash_{LLM} t_i'), p(t_i' \vdash_{LLM} t_i)\} \right). \tag{10}$$

Since the main objective is to find the most plausible proofs, i.e., the sequences of most plausible theory resolution steps, we define the first entry of our priority score for each $c_{res}$ as the overall type consistency score of all resolution steps beginning from $\neg q$ that led to its derivation. Let $\mathcal{P}_{c_{res}}$ be the set of parent clauses of $c_{res}$; the proof type consistency score of $c_{res}$ is inductively defined as

$$\rho^t(c_{res}) = \left( \prod_{c' \in \mathcal{P}_{c_{res}}} \rho^t(c') \right) \cdot \rho^{type}_{c_{res}}. \tag{11}$$

*Resolution Ordering based on Predicate Entailment:* Assigning the type consistency scores prunes the resolution search space ensuring that only clauses with compatible argument types will be considered for resolution. To further enhance efficiency, LLM-TH prioritizes the remaining clauses based on their potential for being part of a plausible proof considering their predicate entailment. As explained, in our LLM-based theory resolution framework, LLM entailment is used to identify unsatisfiability of clauses. Therefore, the greater probability the LLM assigns to $l_{target}$ entailing $l$, the more plausible it becomes to resolve $l$ and $l_{target}$. Therefore we define the plausibility of a theory resolution step between $c$ and $c_{target}$ by resolving literals $l$ and $l_{target}$ generating $c_{res}$, denoted by $\rho^{entail}_{c_{res}}$

$$\rho^{entail}_{c_{res}} = \begin{cases} 1 & \text{if } (l_{target} \vdash l) \in \mathcal{I}, \\ 0 & \text{if } \neg(l_{target} \vdash l) \in \mathcal{F}, \\ p(l_{target} \vdash_{LLM} l) & \text{otherwise .} \end{cases} \tag{12}$$

These plausibility scores can help us prioritize the resolvent clauses based on their predicate entailment. For example, in the scenario depicted in Figure 1, resolving *"Sicilian"* with *"Mediterranean"* results in a higher entailment score compared to resolving *"Mustard-flavored"* with *"Mediterranean"*. Hence, it is prudent to prioritize the former resolvent, as it is more likely to contribute to the final proof. As the definition of the overall proof type consistency score, we can compute the overall predicate entailment score inductively to obtain the second entry of our priority score as

$$\rho^e(c_{res}) = \left( \prod_{c' \in \mathcal{P}_{c_{res}}} \rho^e(c') \right) \cdot \rho^{entail}_{c_{res}}. \tag{13}$$

Ultimately, among equally plausible proofs, we prioritize shorter ones that circumvent unnecessary steps. We define the proof length score, a third priority score that accounts for this preference. The proof length score of $c_{res}$ is derived inductively from the maximum length of its parent clauses as

$$\rho^l(c_{res}) = 1 + \max_{c' \in \mathcal{P}_{c_{res}}} \rho^l(c'). \tag{14}$$

Each resolvent $c_{res}$ is assigned the priority tuple $(\rho^t(c_{res}), \rho^e(c_{res}), \rho^l(c_{res}))$ and then pushed to the priority queue $PQ$, in which the order of clauses is specified as

$$c_1 \preceq c_2 \iff [\rho^t(c_1) > \rho^t(c_2)] \vee [(\rho^t(c_1) = \rho^t(c_2)) \wedge (\rho^e(c_1) > \rho^e(c_2))] \tag{15}$$
$$\vee [(\rho^t(c_1) = \rho^t(c_2)) \wedge (\rho^e(c_1) = \rho^e(c_2)) \wedge (\rho^l(c_1) < \rho^l(c_2))].$$

By applying this prioritization scheme, the type consistency priority score first applies a hard filter to avoid exploration of resolvents with invalid types, and the predicate entailment and length priorities together enable an efficient inference via LLM-based theory resolution. These efficiency enhancements enable LLM-TH to be applied to large-scale KBs. Furthermore, it is able to reason over incomplete KBs by benefiting from the commonsense knowledge of the LLM to fill in the missing axioms by identifying entailing predicates in the theory resolution process.

At the start of each iteration of LLM-TH, the clause with the highest rank in $PQ$ becomes the active clause. When a resolution step yields a contradiction, the proof and its respective priority score are added to the set of found proofs by backtracking the ancestor clauses. The algorithm terminates when either a certain number of proofs are found or the maximum number of iterations is exceeded. Notably, LLM-TH is not limited to proving a single query; instead, it finds a set of proofs and assigns a strength score to each. This feature enables it to evaluate the likelihood of each query being entailed, which is critical for tasks requiring ranking, such as answering multiple-choice questions. Furthermore, *LLM-TH can reason on incomplete KBs by using the LLM's commonsense knowledge to fill in the missing axioms by identifying entailed predicates via theory resolution.*

## 4    Fixing Erroneous Resolutions in LLM-TH

LLM-TH enables verification of the reasoning process by providing access to atomic resolution steps. Therefore, if an incorrect theory resolution step is identified, the source of failure can be provably repaired. In the example of Figure 1, the LLM's mistake in assigning a low probability for *"Souvlaki"* to entail *"Mediterranean"* leads to a missed resolution. Also, incorrectly considering *"cuttlefish"* to entail being a *"fish"* leads to an incorrect resolution. To fix both errors, we (1) insert the missing axiom $\forall y$ *"Souvlaki"*$(y) \implies$ *"Mediterranean"*$(y)$ to $\mathcal{I}$ and (2) insert the forbidden axiom *"cuttlefish"* $\nvdash_{LLM}$ *"fish"* to $\mathcal{F}$. The formal propositions below are proved in Appendix B.

**Proposition 1.** *Consider proof $P_c^\phi$ using axiom $\phi$ that derives clause c. For missing LLM reasoning axiom $\phi$, a Repair Axiom $\phi'$ can be inserted into $\mathcal{I}$ such that $P_c^{\phi'}$ will be produced before $P_c^\phi$.*

**Proposition 2.** *Consider proof $P_c^\phi$ using axiom $\phi$ that derives clause c. For any incorrect LLM reasoning axiom $\phi$, a Repair Axiom $\phi'$ can be inserted into $\mathcal{F}$ such that $P_c^\phi$ will receive priority 0.*

## 5    Experiments

We empirically evaluate LLM-TH[1] on three different tasks representing commonsense reasoning with KBs on different datasets and compare it against variations of four different baselines to compare them from different aspects by answering the following questions:

- **RQ1-Scalability:** How does the reasoning performance of LLM-TH compare to baselines when reasoning with large, but complete knowledge bases?

- **RQ2-Reasoning with incomplete KBs:** How effectively do LLM-TH and the baselines use the LLM's commonsense knowledge to compensate for the incompleteness of the KB?

- **RQ3-Efficiency:** How is the efficiency of LLM-TH influenced by typed hyperresolution?

### 5.1    Tasks and Datasets Description

- **Preference reasoning:** An exemplar commonsense reasoning task is providing recommendations using natural language statements of user preferences and restrictions. For this task, we use Recipe-MPR (Zhang et al., 2023a), a dataset consisting of $500$ queries, e.g., *"I want French food, but I'm on a budget"* and five-way recipe options. This dataset covers a broad range of commonsense reasoning skills such as temporal and analogical reasoning.

- **Multi-domain Deductive reasoning:** Since established datasets for logical commonsense reasoning with LLMs, e.g., ProntoQA (Saparov & He, 2022) and COPA-SSE (Brassard et al., 2022), have small KBs with less than 20 facts and axioms per query, we find them insufficient for evaluating the reasoning capability on large KBs. Thus, following the approach in Saparov & He (2022), we create a deductive reasoning dataset using manually written commonsense axioms and ground facts sampled from Wikidata knowledge graph (Vrandečić & Krötzsch, 2014). This dataset contains more than 32k rules about five different domains: Biological entities, foods, vehicles, drugs and diseases, and sports, and 1000 queries that answering them requires 2 to 7 reasoning steps. We release this dataset to encourage research on LLM-based commonsense reasoning on large-scale KBs.

- **Geographical QA:** Using the same approach for generating the multi-domain deductive reasoning dataset, we create a KB about geographical entities, e.g., cities, deserts, museums, etc. containing more than 12k rules and 500 queries which we also release.

### 5.2    Baselines and Evaluation

We use established methodologies for eliciting more faithful reasoning from the LLMs as our comparison baselines: (a) zero-shot CoT (Kojima et al., 2022), (b) few-shot CoT (Wei et al., 2022), and (c) RAG (Lewis et al., 2020b) using a dense retriever (Song et al., 2020) to find relevant rules from the KB and prompting the LLM with zero-shot CoT and (d) few-shot CoT. (e) To enable comparison to semantic parsing (Pan et al., 2023; Olausson et al., 2023), we provide them with an ideal KB

---

[1]https://github.com/atoroghi/LLM-TH

Table 1: Reasoning performance of methods across the three datasets on complete KBs.

| Method | Preference Reasoning | | | Deductive Reasoning | | | Geographical QA | | |
|---|---|---|---|---|---|---|---|---|---|
| | Accuracy | RS Macro | RS Micro | Accuracy | RS Macro | RS Micro | Accuracy | RS Macro | RS Micro |
| **GPT-3.5 Turbo** | | | | | | | | | |
| Zero-Shot CoT | 0.86±0.04 | 0.60 | 0.80 | 0.69±0.02 | 0.45 | 0.48 | 0.71±0.03 | 0.60 | 0.77 |
| Few-Shot CoT | 0.87±0.02 | 0.65 | 0.81 | 0.65±0.03 | 0.45 | 0.48 | 0.82±0.02 | 0.50 | 0.53 |
| RAG + Zero-Shot CoT | NA | NA | NA | 0.68±0.02 | 0.85 | 0.92 | 0.74±0.01 | 0.80 | 0.88 |
| RAG + Few-Shot CoT | NA | NA | NA | 0.69±0.02 | 0.65 | 0.84 | 0.83±0.02 | 0.75 | 0.84 |
| **Gemini-1.5-Flash** | | | | | | | | | |
| Zero-Shot CoT | 0.84±0.04 | 0.60 | 0.75 | 0.60±0.01 | 0.40 | 0.81 | 0.78±0.03 | 0.20 | 0.51 |
| Few-Shot CoT | 0.86±0.02 | 0.55 | 0.79 | 0.66±0.05 | 0.20 | 0.71 | 0.79±0.02 | 0.25 | 0.54 |
| RAG + Zero-Shot CoT | NA | NA | NA | 0.78±0.02 | 0.85 | 0.93 | 0.79±0.03 | 0.50 | 0.72 |
| RAG + Few-Shot CoT | NA | NA | NA | 0.86±0.04 | 0.45 | 0.72 | 0.78±0.03 | 0.25 | 0.47 |
| **Llama3 70B** | | | | | | | | | |
| Zero-Shot CoT | 0.87±0.01 | 0.55 | 0.80 | 0.80±0.03 | 0.15 | 0.77 | 0.78±0.01 | 0.25 | 0.57 |
| Few-Shot CoT | **0.91±0.01** | 0.70 | 0.84 | 0.78± 0.02 | 0.55 | 0.58 | 0.87±0.02 | 0.45 | 0.45 |
| RAG + Zero-Shot CoT | NA | NA | NA | 0.78±0.01 | 0.50 | 0.81 | 0.87±0.030 | 0.40 | 0.65 |
| RAG + Few-Shot CoT | NA | NA | NA | 0.80±0.02 | 0.75 | 0.80 | 0.91±0.02 | 0.65 | 0.71 |
| **Mixtral 46.7B** | | | | | | | | | |
| Zero-Shot CoT | 0.79±0.03 | 0.60 | 0.84 | 0.59±0.02 | 0.30 | 0.66 | 0.71±0.02 | 0.50 | 0.70 |
| Few-Shot CoT | 0.74±0.02 | 0.65 | 0.83 | 0.67±0.01 | 0.45 | 0.53 | 0.80±0.01 | 0.45 | 0.49 |
| RAG + Zero-Shot CoT | NA | NA | NA | 0.65±0.02 | 0.65 | 0.81 | 0.66±0.07 | 0.25 | 0.51 |
| RAG + Few-Shot CoT | NA | NA | NA | 0.46±0.03 | 0.30 | 0.43 | 0.70±0.06 | 0.50 | 0.65 |
| **VERA** (T5 5B) | 0.86 | NA | NA | 0.76 | NA | NA | 0.68 | NA | NA |
| **Semantic Parsing** | NA | NA | NA | NA | NA | NA | 0.05 | NA | NA |
| **LLM-TH** (BART 406M) | 0.84 | **0.90** | **0.94** | **1.00** | **1.00** | **1.00** | **1.00** | **1.00** | **1.00** |

including all axioms necessary to answer a perfect semantic parse of the query (cf. Appendix C). (e) We also compare against (f) VERA (Liu et al., 2023), a method for estimating commonsense plausibility of statements. VERA estimates a score in the range $(0, 1)$, and we consider a score greater (smaller) than $0.5$ as a True (False) prediction. Except VERA which uses T5 (Raffel et al., 2020), we use different common LLMs for other baselines: (1) Gemini 1.5-flash, (2) Llama3 (70B), (3) Mixtral (56.7B), and (4) GPT3.5 Turbo, while using BART large (Lewis et al., 2020a)[2] (406 M) and RoBERTa large (Liu et al., 2019)[3] (365 M) tuned on MNLI (Williams et al., 2018) dataset to obtain entailment probabilities for LLM-TH. We use pyDatalog[4] to perform hyperresolution for grounding on the KB facts and use Gemini 1.5-flash to convert natural language axioms to clausal form.

We evaluate the reasoning performance based on both, (1) the correctness of the final answer, measured by *accuracy*, and (2) the correctness of the reasoning process by evaluating proofs using the *reasoning score* (RS) (Kazemi et al., 2023) metric which is manually calculated for 20 randomly chosen responses in which the final answer was correct. RS is typically assessed as a binary decision based on whether the predicted proof is entirely aligned with the ground truth proof (Kazemi et al., 2023; Lee & Hwang, 2024), which leads to both a single erroneous step and wholly flawed reasoning being penalized equally. To achieve a more granular evaluation of the proofs, we calculate both the conventional *macro RS* and following the idea of Min et al. (2023), we propose a metric which we call *micro RS*. Let $P$ be a provided proof and $P^*$ a valid ground truth proof. Using the indicator function $\mathbb{I}$, we define the micro RS for each query as $RS_{Micro} = \frac{1}{|P|} \sum_{p \in P} \mathbb{I}(p \in P^*)$.

Table 2: Reasoning performance on incomplete KBs. Numbers in parenthesis indicate the difference with the method's performance on a complete KB (Table 1) with ↓ (↑) showing a decrease (increase).

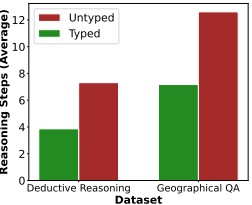

Figure 3: Influence of typing on the efficiency of the inference algorithm.

| Method | Deductive Reasoning | | Geographical QA | |
|---|---|---|---|---|
| | Zero-Shot CoT | Few-Shot CoT | Zero-Shot CoT | Few-Shot CoT |
| GPT-3.5-Turbo | 0.65±0.01 (0.03↓) | 0.54±0.04 (0.15↓) | 0.60±0.02 (0.14↓) | 0.64±0.03 (0.19↓) |
| Gemini-1.5-Flash | 0.73±0.02 (0.05↓) | 0.73±0.01 (0.13↓) | 0.68±0.02 (0.11↓) | 0.66±0.02 (0.11↓) |
| Llama3 70B | 0.77±0.02 (0.01↓) | 0.77±0.01 (0.03↓) | 0.66±0.02 (0.21↓) | 0.66±0.02 (0.25↓) |
| Mixtral 46.7B | 0.50±0.04 (0.14↓) | 0.48±0.02 (0.03↑) | 0.53±0.03 (0.12↓) | 0.58±0.03 (0.13↓) |
| LLM-TH (BART 406M) | **0.97** | - | **0.95** | - |
| LLM-TH (RoBERTa 356M) | **0.96** | - | **0.92** | - |

[2] https://huggingface.co/facebook/bart-large-mnli

[3] https://huggingface.co/FacebookAI/roberta-large-mnli

[4] https://pypi.org/project/pyDatalog/

**5.2.1 RQ1: Reasoning Performance on Complete KB**  Before we proceed to the general case of reasoning with incomplete KBs, we first aim to study whether the typed hyperresolution methodology of LLM-TH can scale to answer queries in an ideal and complete, yet large KB in comparison to the baselines. These results serve as an upper bound for incomplete KB reasoning performance and also allow us to distinguish failure modes related to reasoning from failure modes related to KB incompleteness assessed in subsequent experiments. As the results in Table 1 indicate, on the preference reasoning task with a small and implicit KB of the five available recipe options, although zero-shot and few-shot CoT with a large LLM such as Llama 3 (70B) yield superior accuracy, LLM-TH still shows a reasonable performance, exceeding zero-shot and few-shot CoT with Mixtral and is competitive with zero-shot CoT using GPT3.5 and Gemini despite using a substantially smaller LLM. For the reasoning scores, LLM-TH exhibits a more correct and faithful reasoning process than all other methods. On this task, since the dataset does not contain an explicit KB, RAG-based baselines reduce to zero-shot and few-shot CoT. On larger KBs of deductive reasoning and geographical QA tasks, the limitations of existing LLM-based methods are revealed as none of them compare to the accuracy of LLM-TH. Furthermore, they all obtain imperfect reasoning scores, reflecting their hallucinations and reasoning errors. On these datasets, LLM-TH performs standard resolution which is a sound and complete inference rule, as reflected in the results. However, the existing semantic parsing methods that use off-the-shelf theorem provers (intended for mathematical theorem proving) are not scalable to these large instance-heavy KBs, even with an ideal parser. While complete KBs are impractical in real-world use cases, results of this experiment highlight that existing baselines, as opposed to LLM-TH, exhibit reasoning failures on large scale KBs *even* when they are complete.

**5.2.2 RQ2: Reasoning Performance on Incomplete KBs**  To enable reasoning over practical KBs, leveraging the commonsense reasoning ability of the LLM to compensate for KB incompleteness is essential. To assess this capability, in our experiments on deductive reasoning and geographical QA datasets that have explicit KBs, we simulate KB incompleteness by omitting one of the rules used in the proof of each query, to test whether the LLM can use its commonsense knowledge to deduce, e.g., *"paying taxes"* implies *"earning revenue"*. Since few-shot and zero-shot CoT methods rely solely on the internal LLM knowledge, they are irrelevant to this RQ that examines the role of KB incompleteness. Hence, we compare LLM-TH against variations of RAG with zero-shot and few-shot CoT prompting. Results in Table 2 show that using the theory hyperresolution framework, LLM-TH is able to achieve significantly higher accuracies compared to the RAG-based baselines that clearly struggle with incompleteness compared to RAG results for complete KBs in Table 1.

In order to provide further insight into the failure cases of RAG-based methods and the limitations of LLM-TH, we conduct a series of additional experiments with RAG-based methods using an ideal retriever, as well as robustness of different methods to longer reasoning chains and different levels of KB incompleteness in Appendix D. We provide comparative anecdotal examples in Appendix H.

**5.2.3 RQ3: Influence of Typing on Efficiency**  To verify the efficiency enhancement offered by introducing type information to the hyperresolution framework of LLM-TH, we perform an ablation experiment on the complete KB, by comparing the average number of reasoning steps that LLM-TH takes to find the answers with its untyped variant that does not consider types in prioritization. In summary, the results of this experiment shown in Figure 3 , indicate that using the typed hyperresolution framework, a considerable improvement is obtained in the efficiency of the reasoning process by reducing the number of reasoning steps required to find the answers to almost half compare to the untyped variant. This efficiency benefit is crucial in interacting with vast real-world KBs.

## 6 Conclusion

We proposed LLM-TH for logical commonsense reasoning with large and incomplete KBs. Using theory resolution, LLM-TH integrates LLM commonsense knowledge into the resolution inference rule to enable reasoning over incomplete KBs with arbitrary predicates. LLM-TH shows strong performance: it matches or outperforms baselines that use orders of magnitude larger LLMs; its use of an LLM-based typed hyperresolution approach yields high efficiency gains; and its transparency and repairability establish it as a solution for factual and correct reasoning on large-scale KBs. In summary, LLM-TH holds promise to significantly reduce hallucinations in LLM-based reasoning.

**Acknowledgments**   This work was supported by LG Electronics, Toronto AI Lab Grant Ref No. 2024-0565. We also thank Roxana Li for her valuable assistance in evaluations.

**Ethics Statement:**   By introducing LLM-TH, we tried to enhance the transparency and increasing control over the reasoning process of LLM-based logical commonsense reasoning. However, drawing logically valid conclusions does not necessarily mean that all axioms, rules, and the internal commonsense knowledge of the LLM which are leveraged in the reasoning process follow ethical requirements. A responsible and credible usage of LLM-TH, like any other reasoning framework, requires careful considerations and assessments of the knowledge base, the underlying LLM, and the user-defined axioms to ensure desired unbiased and ethical performance.

**Reproducibility Statement:**   We release all our code and data in the supplementary materials and the LLM-TH repository[5]. We also explain the experimental setup and dataset descriptions in Section 5, and include all prompts utilized for the LLM usage in Appendix G, as well as in the supplementary materials.

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

## A    PROOF OF LEMMA 1

**Lemma 1.** *Resolving two disjunctive clauses $c_1$ and $c_2$ that include complimentary literals $l(x_1, ..., x_n)|x_1 : T_1, ..., x_n : T_n$ and $\neg l(y_1, ..., y_n)|y_1 : H_1, ..., y_n : H_n$ under the unifier $\theta = \{x_1/y_1, ..., x_2/y_2\}$ creates a resolvent $c_{res}$ with type variables $x_1 : T_1 \sqcap H_1, ..., x_n : T_n \sqcap T_n$.*

*Proof.* Assume $c_1$ to be $A(x_1, ..., x_n) \vee l(x_1, ..., x_n)|x_1 : T_1, ..., x_n : T_n$ and $c_2$ to be $\neg l(y_1, ..., y_n) \vee B(y_1, ..., y_n)|y_1 : H_1, ..., y_n : H_n$. Following 5 and 6, we can rewrite these clauses in implication form FOL by adding type predicates as

$$\forall x_1, ..., \forall x_n \bigwedge_{i=1}^{n} T_i(x_i) \wedge \neg A(x_1, ..., x_n) \implies l(x_1, ..., x_n), \tag{16}$$

$$\forall y_1, ..., \forall y_n \bigwedge_{i=1}^{n} H_i(y_i) \wedge l(y_1, ..., y_n) \implies B(y_1, ..., y_n), \tag{17}$$

Converting these clauses to the disjunctive form yields

$$\forall x_1, ..., \forall x_n \bigvee_{i=1}^{n} \neg T_i(x_i) \vee A(x_1, ..., x_n) \vee l(x_1, ..., x_n), \tag{18}$$

$$\forall y_1, ..., \forall y_n \bigvee_{i=1}^{n} \neg H_i(y_i) \vee \neg l(y_1, ..., y_n) \vee B(y_1, ..., y_n). \tag{19}$$

Now, we can resolve $l(x_1, ..., x_n)$ with $\neg l(y_1, ..., y_n)$ under the unifier $\theta = \{x_1/y_1, ..., x_2/y_2\}$ as

$$\frac{\bigvee_{i=1}^{n} \neg T_i(x_i) \vee A(x_1, ..., x_n) \vee l(x_1, ..., x_n) \quad \bigvee_{i=1}^{n} \neg H_i(x_i) \vee \neg l(y_1, ..., y_n) \vee B(y_1, ..., y_n)}{\bigvee_{i=1}^{n} \neg T_i(x_i) \bigvee_{i=1}^{n} \neg H_i(x_i) \vee A(x_1, ..., x_n) \vee B(x_1, ..., x_n)}, \tag{20}$$

which can be rewritten as

$$\forall x_1, ..., \forall x_n \bigwedge_{i=1}^{n} T_i(x_i) \wedge H_i(x_i) \implies A(x_1, ..., x_n) \vee B(x_1, ..., x_n). \tag{21}$$

Therefore, the unary type predicates for each $x_i$ becomes the conjunction of the types from their parent clauses, which in our typed FOL notation, can be equivalently written as

$$\forall x_1, ..., \forall x_n A(x_1, ..., x_n) \vee B(x_1, ..., x_n)|x_1 : T_1 \sqcap H_1, ..., x_n : T_n \sqcap H_n. \tag{22}$$

$\square$

## B    PROOF OF REPAIRABILITY OF LLM-TH

Recall from Section 3.1 that we have two distinct sets of a set of repair axioms:

- **Intended repair axioms $\mathcal{I}$:** these axioms are used to provide missing information to the KB and are assigned priority score 1 (cf. Equation 12) during inference since they are user-provided and known to be true. A proof with a repair axiom will always receive higher priority over a proof using an equivalent LLM inference (during theory resolution), since the equivalent LLM inference from entailment necessarily has priority score $< 1$.

- **Forbidden repair axioms $\mathcal{F}$:** these axioms are intended to override erroneous LLM inferences (during theory resolution) and are assigned priority 0 (cf. Equation 12) during inference. Any proof that uses such a forbidden axiom will automatically receive priority score 0 due to Equation 13.

We now formalize and prove these statements of correctness. We begin by defining a proof $P_c^{\phi} = P_c \cup \{\phi\}$ as the combined set of clauses $P_c$ and $\phi$ that derive clause $c$.

**Proposition 1.** *Consider proof $P_c^\phi$ using axiom $\phi$ that derives clause c. For missing LLM reasoning axiom $\phi$, a Repair Axiom $\phi'$ can be inserted into $\mathcal{I}$ such that $P_c^{\phi'}$ will be produced before $P_c^\phi$.*

*Proof.* We can obtain the proof score $\rho^e(P_c^\phi)$ of clause $c$ by inductively unrolling Equation 13 for $\rho^e(c)$ over all ancestor clauses $P_c^\phi$ that derive it. This yields a simple product form: $\rho^e(P_c^\phi) = \rho_\phi^{entail} \cdot \prod_{c' \in P_c} \rho_{c'}^{entail}$. Now, comparing two different derivations $P_c^\phi$ and $P_c^{\phi'}$ of $c$, we can easily show that $\rho^e(P_c^{\phi'}) > \rho^e(P_c^\phi)$ since $\frac{\rho^e(P_c^{\phi'})}{\rho^e(P_c^\phi))} = \frac{\rho_{\phi'}^{entail} \cdot \prod_{c' \in P_c} \rho_{c'}^{entail}}{\rho_\phi^{entail} \cdot \prod_{c' \in P_c} \rho_{c'}^{entail}} = \frac{\rho_{\phi'}^{entail}}{\rho_\phi^{entail}} > 1$ given that the explicit Repair Axiom has $\rho_{\phi'}^{entail} = 1$ (following Equation 13) while the LLM entailment score $\rho_\phi^{entail} < 1$. Hence, the proof $P_c^{\phi'}$ containing the Repair Axiom $\phi'$ will always be given precedence over $P_c^\phi$ according to the total ordering of Equation 15 used to prioritize proofs in the LLM-TH Algorithm 1. □

**Proposition 2.** *Consider proof $P_c^\phi$ using axiom $\phi$ that derives clause c. For any incorrect LLM reasoning axiom $\phi$, a Repair Axiom $\phi'$ can be inserted into $\mathcal{F}$ such that $P_c^\phi$ will receive priority 0.*

*Proof.* In the presence of $\phi'$, the priority score assigned to the resolvent clause $c$, i.e., $\rho_\phi^{entail}$ will be set to 0 following Equations 12, which then implies that the priority score of proof $P_c^\phi$ must be 0 according to 13), which is the product of priority scores of all constituent clauses of the proof. □

In general, priority 0 proofs contain a forbidden or zero probability entailment and should be avoided at inference time.

## C   COMPARISON TO EXISTING LLM-BASED FORMAL REASONING BASELINES

Existing formal reasoning methods with LLMs, i.e., semantic parsing methods and methods emulating inference rules, suffer from two limitations that make them inapplicable to our studied datasets: (i) they cannot leverage the internal commonsense knowledge of the LLM and only rely on an explicit and complete rule base, which Recipe-MPR lacks. (ii) They are limited to small KBs that fit in the LLM context size, but our studied KBs are much larger.

Semantic parsing methods use the LLM to translate the KB and query to symbolic forms. Although these methods use the LLM solely for translating the NL problem into symbolic format, they include all facts and axioms to the LLM in the context because they need the LLM to use consistent predicates and objects in its symbolic translation (otherwise, the symbolic reasoner will not be able to execute the symbolic problem). in a recent work, Xu et al. (2024) showed that even by including all facts and axioms in the context, these methods can have very high execution failure rates due to translation inconsistencies.

Since our large KBs exceed the LLM context limit, we cannot use the LLM in the same manner that semantic parsing baselines, Logic-LM (Pan et al., 2023) and LINC (Olausson et al., 2023), use it for the translation process. Therefore, to ensure a fair comparison, we manually converted all of our KB and queries to the symbolic format supported by Prover9 [6], the FOL theorem prover that both of these works used. We set the time limit of Prover9 to the maximum time it takes for LLM-TH to answer the same queries—as it does not support specifying the maximum number of inference steps. The major difference between LINC and Logic-LM is that the former performs multiple translations and uses majority voting on their results, but since we are using a perfect KB, they become equivalent. Similar to Logic-LM, we also experiment with two backup mechanisms for cases in which the symbolic solver fails: (i) randomly guessing the answer and (ii) using CoT.

When experimenting on our deductive reasoning dataset (KB with about 32k rules), the Prover9 solver crashed and could not reason on problems with such large KB sizes. For the geographical QA dataset (KB with about 12k rules), the following results were obtained:

| Method | Accuracy |
|---|---|
| Logic-LM / LINC with Perfect Translation (no backup mechanism) | 0.05 |
| Logic-LM / LINC with Perfect Translation and random guess backup | 0.61 |
| Logic-LM / LINC with Perfect Translation and CoT backup (GPT3.5) | 0.73 |

Table 3: Accuracies obtained by semantic parsing baselines on the geographical reasoning data.

The underlying cause of Logic-LM's and LINC's imperfect symbolic reasoning component on these complete KBs is due to a lack of design and optimization for large KBs with thousands of facts. To support this claim, it is important to note that the problems used by Logic-LM [2] and LINC [3] have much smaller KBs—often smaller than 30 facts—for which existing theorem provers like Prover9 are sufficient as their experiments show. However, LLM-TH is equipped with efficiency enhancement and prioritization mechanisms such as typing and hyperresolution that significantly improve reasoning efficiency in these settings.

---

[6] https://www.cs.unm.edu/ mccune/prover9/

## D    ADDITIONAL EXPERIMENTS

In this section, we provide a set of additional experiments for a more detailed evaluation of the LLM-TH and the studied baselines. Since the incomplete KB setting is more common in practical use cases, studying the behavior of different models in this setting is more important than the complete KB case (which was studied in Section 5.2.1), all additional experiments that are introduced in this section focus on the incomplete KB case.

### D.1    PERFORMANCE OF RAG-BASED BASELINES WITH IDEAL RETRIEVAL

In section 5.2.2, we studied the performance of different RAG-based baselines on the incomplete KB cases on the deductive reasoning and geographical QA cases and showed that LLM-TH considerably outperforms these baselines. However, it is not clear whether the inferior performance of RAG-based baselines is due to the failure of the retriever to retrieve the relevant facts and axioms, or that the LLMs are challenged with indicating the missing commonsense axioms from an incomplete KB and compensating for them. In this experiment, to have a more fine-grained analysis of the performance of the RAG-based baselines, we repeat the experiments with these baselines, but this time we ensure that all the relevant facts and axioms in the ground truth proof are included in the set of retrieved rules to simulate an *ideal retrieval* setting.

Results of this experiment are presented in Table 4. As expected, in all cases, the performance of different methods with ideal retrieval is enhanced considerably, which indicates that the retriever failure is a critical limitation in the performance of the RAG-based baselines. However, even though under the ideal retrieval condition, the gap between the performance of the LLM-TH and that of the RAG-based baselines is narrowed, e.g., Llama3 70B and Gemini 1.5-Flash achieve accuracies of 0.94 and 0.93 with ideal retrieval and few-shot prompting, the accuracies obtained by LLM-TH using considerably smaller LLMs is still superior. This experiment indicates the restrictions of directly applying LLMs to the task of logical reasoning on incomplete KBs, as they have limited ability to correctly identify the missing commonsense axioms to compensate for the KB incompleteness and provide logical proof, a task that LLM-TH can perform more aptly.

Table 4:   Comparison of the performance of RAG-based baselines with an ideal retriever with the original retriever used. The substantial improvement in results when using an ideal retriever suggests a notable part of the incorrect answers are due to failure in the retrieval stage. However, even by using an ideal retriever, the RAG-based baselines cannot outperform LLM-TH, despite using substantially larger LLMs. These results indicate that existing LLM-based methods are challenged with the task of reasoning over incomplete KBs, whereas LLM-TH can better identify the missing axioms and compensate for them.

| Method | Deductive Reasoning | | | | Geographical QA | | | |
| | Zero-Shot CoT | | Few-Shot CoT | | Zero-Shot CoT | | Few-Shot CoT | |
| | Ideal Retrieval | Real Retrieval | Ideal Retrieval | Real Retrieval | Ideal Retrieval | Real Retrieval | Ideal Retrieval | Real Retrieval |
|---|---|---|---|---|---|---|---|---|
| GPT-3.5-Turbo | 0.73 | 0.65 | 0.67 | 0.54 | 0.64 | 0.60 | 0.69 | 0.64 |
| Gemini-1.5-Flash | 0.85 | 0.73 | 0.93 | 0.73 | 0.86 | 0.68 | 0.86 | 0.66 |
| Llama3 70B | 0.87 | 0.77 | 0.94 | 0.77 | 0.75 | 0.66 | 0.89 | 0.66 |
| Mixtral 46.7B | 0.67 | 0.50 | 0.71 | 0.48 | 0.70 | 0.53 | 0.68 | 0.58 |
| LLM-TH (using BART 406M) | - | **0.97** | - | - | - | **0.95** | - | - |
| LLM-TH (using RoBERTa 356M) | - | **0.96** | - | - | - | **0.92** | - | - |

### D.2    ROBUSTNESS OF DIFFERENT METHODS TO LONGER REASONING CHAINS

As explained in Section 5.1, our deductive reasoning and geographical QA datasets include queries that answering them requires different number of reasoning steps. Concretely, the deductive reasoning dataset contains queries whose reasoning lengths range from 2 to 7, and the geographical QA dataset contains queries of length 2, 8, and 11. In this experiment, we want to study the robustness of LLM-TH and the RAG-based baselines against increase in the length of reasoning steps required to answer queries.

To this end, we provide a fine-grained analysis of the results provided in Table 2 by evaluating the accuracy obtained by each method for each group of queries that require equal number of reasoning steps. Results of this experiment for the deductive reasoning and geographical QA datasets are provided in Figure 4 and 5 respectively. For clarity of representation, we provide results of zero-shot prompting and few-shot prompting with the RAG baselines in separate plots. These results show that LLM-TH maintains its superior performance over the baselines across queries with different numbers of reasoning steps in most cases. Overall, LLM-TH and most baselines show a proper robustness to an increase in the number of reasoning steps. For instance, in the deductive reasoning dataset, most methods (including LLM-TH) obtain lower accuracy for queries with a reasoning length of 3, compared to the queries with a reasoning length of 4. This suggests that in reasoning over the incomplete KB, the difficulty incurred by an increase in the proof length is dominated by the difficulty of commonsense reasoning required to relate the query predicates to the correct predicates from the KB. For instance, finding the connection between the query predicate *"Sold by businesses in"* to the KB axiom predicate *"Consumed by people in"* (a query about foods with a proof length of 3) is more challenging for these methods than finding the connection between *"Can be used interchangeably"* and *"Is prescribed for same condition"* (a query about medications with a proof length of 4).

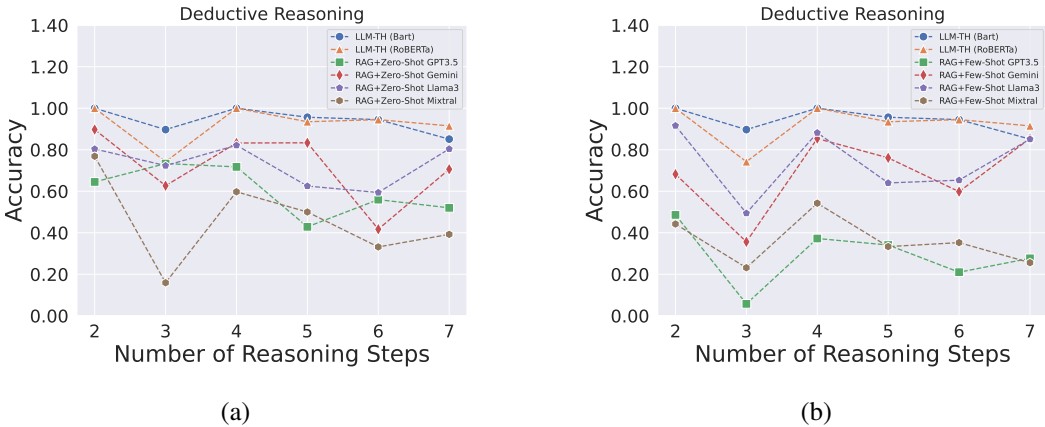

(a)              (b)

Figure 4: Study of robustness of different methods to an increase in the length of reasoning chains required to answering queries from the deductive reasoning dataset. Plot (a) compares LLM-TH to RAG-based baselines with zero-shot prompting and plot (b) compares it to RAG-based baselines with few-shot prompting.

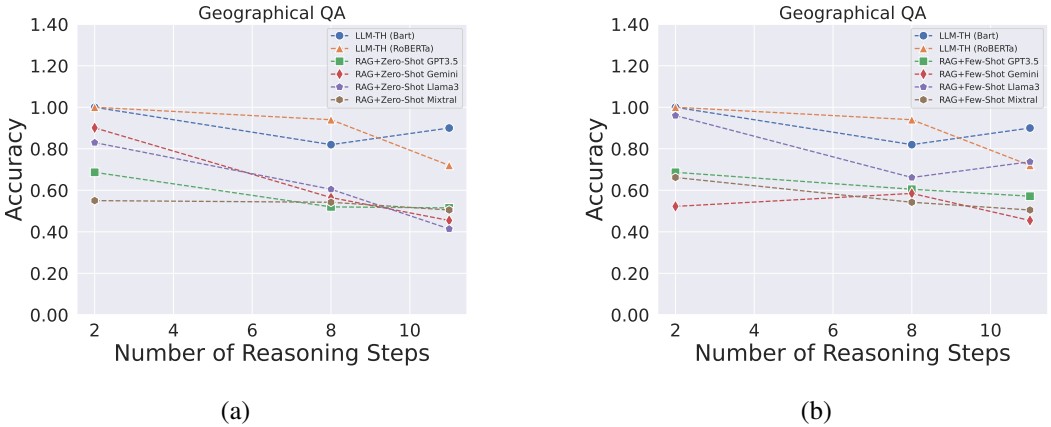

(a)              (b)

Figure 5: Study of robustness of different methods to an increase in the length of reasoning chains required to answering queries from the geographical QA dataset. Plot (a) compares LLM-TH to RAG-based baselines with zero-shot prompting and plot (b) compares it to RAG-based baselines with few-shot prompting.

### D.3 ROBUSTNESS OF DIFFERENT METHODS TO DIFFERENT LEVELS OF KB INCOMPLETENESS

In our previous experiments on the incomplete KB case, we ablated one commonsense axiom to ensure no direct connection exists between the query and the KB axioms. However, practical KBs may suffer from various levels of incompleteness, therefore reasoning on them may require performing multiple steps of commonsense reasoning to find the connection between the query and KB rules.

In this experiment, we study the robustness of LLM-TH and the RAG-based baselines against different levels of KB incompleteness. To this end, we ablate different number of commonsense axioms from the KB and perform experiments using each method. Considering the number of KB axioms used in the ground truth proof for each query, the number of rules we can ablate is different. Therefore, we group queries based on the maximum number of rules that can be ablated, while still a valid proof can be found for it by using KB facts and axioms.

Results of this experiment for the deductive reasoning and geographical QA datasets are provided in Figures 6 and 7 respectively. As expected, with an increase in the number of ablated rules, the commonsense reasoning required to find the connection between the query predicate and the remaining commonsense axioms from the KB axioms becomes more challenging, resulting in a drop in the performance of most methods. In particular, we observe that the performance of LLM-TH using standard NLI models (both BART and RoBERTa) drops drastically with an increase in the number of required reasoning steps, whereas RAG-based baselines exhibit a stronger robustness to the KB incompleteness.

This trend stems from the limitation of entailment models in performing multi-step reasoning which is required to answer queries with an increased level of KB incompleteness. For instance, when only 1 commonsense axiom is ablated from the KB, finding the proof requires theory resolving the query predicate *"Submitted tax forms in"* with the predicate *"Earned revenue in"* from the KB which the entailment models are capable of. However, when 3 rules are ablated, the required theory resolution is between *"Submitted tax forms in"* with *"Played in"* which requires multiple steps of reasoning (e.g., *"since the person played in a professional sport club which is located in a country, they have earned income and therefore should have paid taxes in that country which requires submitting tax forms"*).

This multi-step reasoning is beyond the capability of relatively small NLI models that are employed by LLM-TH. To overcome this limitation, we also experiment LLM-TH by using a larger LLM, Llama 3.1 (8B). Since Llama 3.1 does not directly provide entailment probabilities, we study two different methods to obtain entailment probabilities between two predicates: (i) we provide Chain of thought prompting with few-shot examples, asking the LLM to provide an entailment score between 0 to 1, and (ii) we obtain the joint log probabilities of the sequence *"predicate1 entails predicate 2."*. As shown in Figure 4, using the log probability method still does not provide a satisfactory performance, since this approach also does not offer the required multi-step reasoning. However, using the chain of thought prompting approach, allows the LLM to perform the multi-step reasoning and eventually propose a final entailment score which results to a robust performance of LLM-TH in even higher levels of KB incompleteness, outperforming RAG-based baselines that use even larger LLMs.

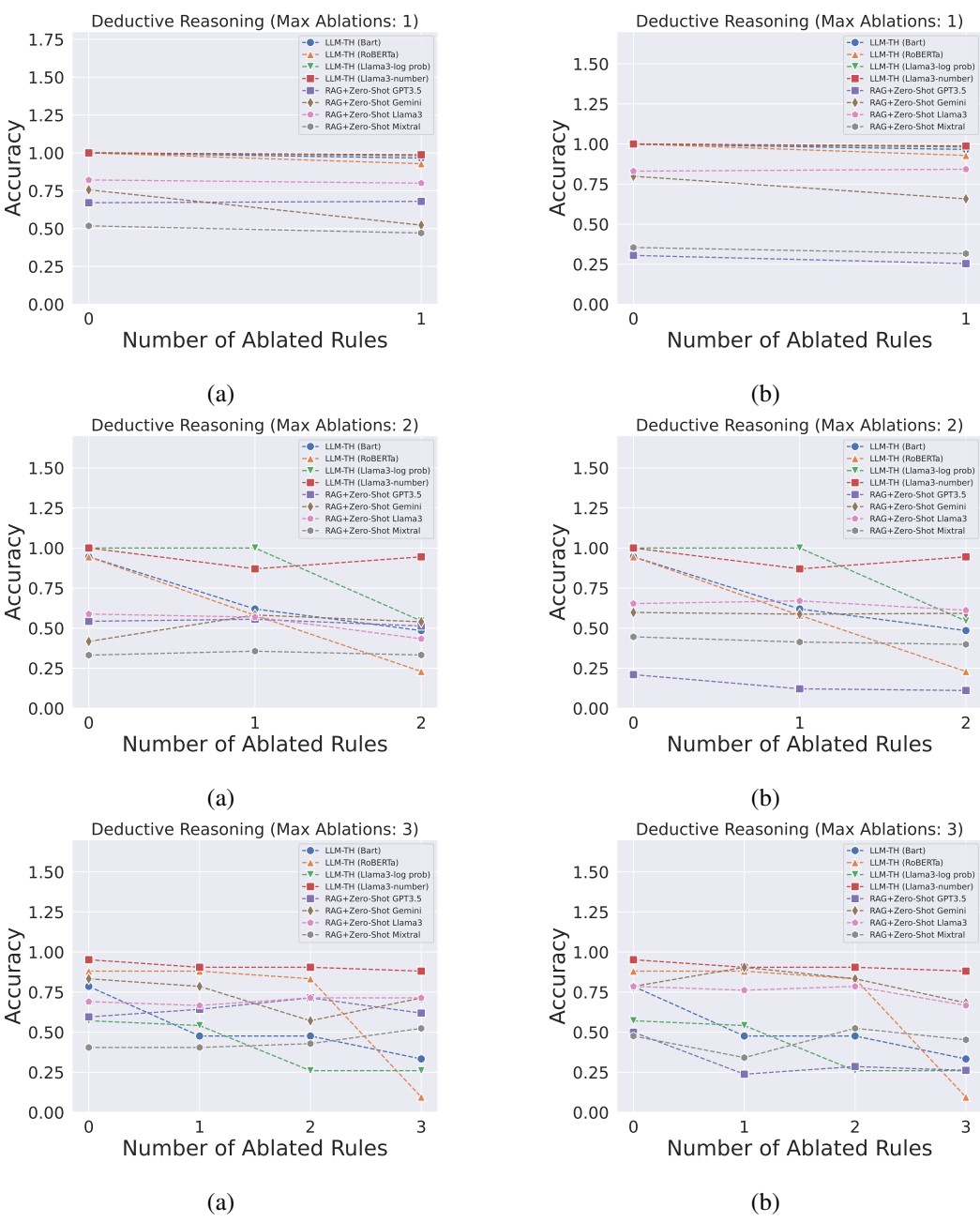

Figure 6: Study of robustness of different methods to different levels of KB incompleteness on the deductive reasoning dataset. Due to the limitation of NLI models in multi-step reasoning, performance of LLM-TH drops when multiple axioms are ablated from the KB. This restriction is addressed when Llama 3.1 (8B) with CoT prompting and few-shot examples is used to obtain the entailment scores, resulting in a robust performance from LLM-TH.s

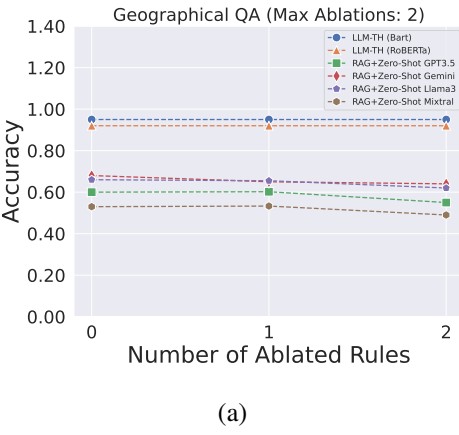
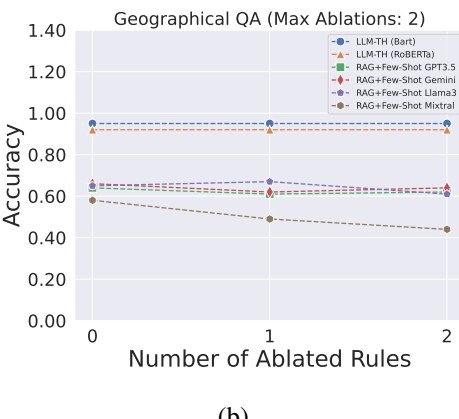

|  |  |
|:---:|:---:|
| (a) | (b) |

Figure 7: Study of robustness of different methods to different levels of KB incompleteness on the geographical QA dataset. LLM-TH shows a more robust performance to ablation of commonsense axioms in this task, compared to deductive reasoning.

# E  EXPLANATION OF THE LLM-TH ALGORITHM

In this section, we provide a detailed explanation of the LLM-TH algorithm, presented in Algorithm 1.

The algorithm takes as inputs the knowledge base ($\mathcal{K}$), the query ($q$), a computational budget defined by the maximum number of iterations (*max_iters*), the maximum number of found proofs shown by *max_proofs*, and two optional sets of repair axioms: (i) the set of intended resolutions denoted by $\mathcal{I}$, and (ii) the set of forbidden resolutions shown by $\mathcal{F}$.

As explained in Section 3.1, LLM-TH is not limited to finding a single proof to the query, but rather it can find a number of proofs with different plausibilities, a capability that is necessary for answering multiple-choice questions. For example, in the preference reasoning example provided in Figure 1, it finds a proof for each of the candidate recipes with a plausibility score assigned to each proof, which is used to rank the recipe options.

The algorithm initializes an empty set, *proofs*, to store discovered proofs, an iteration counter $i$, and an initially empty priority queue $PQ$. $PQ$ stores resolvent clauses and orders them according to their priority scores, using Equation 15. In the backward chaining approach used in LLM-TH, the first clause stored in $PQ$ is $\neg q$ with the priority of $(1, 1, 0)$. The algorithm's main loop executes as long as the priority queue ($PQ$) is not empty, the maximum proof limit is not reached, and the iteration cap is not exceeded. During each iteration, the clause $c$ with the highest priority in $PQ$ is dequeued and designated as the active clause. In the initial iteration, since $PQ$ contains only $\neg q$, this clause becomes the first active clause.

If the active clause is $\bot$, the previous resolution step has resulted in a contradiction. The proof can then be reconstructed by tracing the ancestors of $c$, starting from its parent set $\mathcal{P}_c$ and continuing up to the initial clause $\neg q$. Once a proof is found, it is added to the set *proofs*. However, if $c \neq \bot$, the algorithm proceeds by selecting clauses to resolve with $c$. It first identifies $\beta_c$, the set of candidate clauses in $\mathcal{K}$ that are resolvable with $c$. A clause must meet two conditions to be resolvable with $c$ in a theory resolution step following Equation 3: (i) it must share the same arity as $c$, and (ii) it must have a different polarity from $c$. After forming $\beta_c$, LLM-TH iterates through all clauses $c_{\text{target}} \in \beta_c$ and resolves each with $c$ via hyperresolution, as described in Equation 2, to obtain $c_{\text{res}}$. It then forms the priority tuple using Equations 11 , 13 , and 14 , and adds $c_{\text{res}}$ to $PQ$ based on this priority.

The algorithm continues until either the maximum number of proofs is found, $PQ$ becomes empty, or the maximum allowed number of iterations is reached.

## F   COMMONSENSE REASONING WITH LLMS

The general conception that humans have about the world and how it works, often referred to as commonsense knowledge, and the ability to reason about it, known as commonsense reasoning, are essential capabilities that average human beings possess. In order to be able to interact with the real world and humans, AI agents are also required to obtain this knowledge (Baroni et al., 2017; Shen & Kejriwal, 2023). Therefore, commonsense reasoning has been a central problem in AI over the past decades (Moore, 1982; Liu & Singh, 2004; Davis & Marcus, 2015). With the advent of language models, and particularly, transformer-based architectures, models were developed that exhibited a decent performance on some commonsense reasoning tasks such as physical and social reasoning (Bisk et al., 2020; Sap et al., 2019). Further improvements were obtained by scaled language models, i.e., LLMs, as they exhibited an outstanding capability of commonsense understanding and reasoning across various evaluation benchmarks (Zhao et al., 2024; Krause & Stolzenburg, 2023; Guo et al., 2024).

These improvements paved the way for inspecting new aspects of commonsense reasoning, such as reasoning on problems involving logical inference, a vein of work referred to as *complex commonsense reasoning* (Ismayilzada et al., 2023; Singh et al., 2021; He et al., 2021). The introduction of this line of work shed light on important limitations of LLMs in challenging commonsense reasoning tasks. Although LLMs have shown outstanding reasoning performance on simple commonsense reasoning benchmarks, they struggle with more challenging tasks, such as commonsense reasoning in real-world use cases (Paul et al.).

These remaining challenges and the risk of reasoning errors restrict the deployability of LLMs in real-world applications, especially in high-stakes scenarios. Furthermore, LLMs lack the factual knowledge required to reason in various applications, such as information about recent events or personal data (Toroghi et al., 2024b). In these applications, factual KBs such as knowledge graphs have been widely exploited to represent factual knowledge in applications ranging from healthcare (Rastogi & Zaki, 2020) to recommendation (Toroghi et al., 2023; Raza et al., 2024; Toroghi & Sanner, 2024). Despite the extensive amount of existing work on commonsense reasoning with LLMs, to the best of our knowledge, no work has targeted the problem of combining commonsense reasoning with reasoning over factual KBs. This problem is particularly important since LLMs show factual hallucinations when they lack domain-specific knowledge (Huang et al., 2023; Tonmoy et al., 2024). In this work, we aim to take complex logical commonsense reasoning task a step further, by proposing a methodology that leverages both commonsense reasoning knowledge of the LLM, as well as the factual knowledge of the KB.

## G   PROMPTS USED FOR LANGUAGE MODELS

We provide the prompts that we used for the LLMs in the experiments of this paper. They are also included in our repository along with the implementation code and data.

**Prompt for Preference Reasoning Task (Zero-shot and Few-shot CoT)**

```
Task: You will be given a query that asks for a recipe and five
options that you have to choose from. Think step by step. First
state your reason for your choice and then say: "Therefore, the
selected recipe is <recipe id>.
Query: {{QUERY}}
[Examples if few-shot:]
```

**Prompt for Multi-domain Deductive Reasoning Task (Zero-shot and Few-shot CoT)**

```
Task: You will be given a query about some knowledge graph
entities in the form of a first order logic predicate that is
either True or False (for example, "CanHoldIn(Apple, Basket)"
which means one can hold an apple in a basket). Your task is to
identify whether the answer to this query is "True" or "False"
and also provide a proof of the answer. First, state your proof
mentioning the rules you used and then say: "Therefore, the
```

```
answer is True" or "Therefore, the answer is False". Think step
by step.
Query: {{QUERY}}
[Examples if few-shot:]
```

**Prompt for Multi-domain Deductive Reasoning (RAG with Zero-shot and Few-shot CoT)**

```
Task: You will be given a query about some knowledge graph
entities in the form of a first order logic predicate that is
either True or False (for example, "CanHoldIn(Apple, Basket)"
which means one can hold an apple in a basket) and a Knowledge
Base containing a set of rules that will help you identify the
answer. Your task is to identify whether the answer to the query
is "True" or "False" and also provide a proof of the query using
the knowledge base. First state your proof mentioning the rules
you used and then say: "Therefore, the answer is True" or
"Therefore, the answer is False". Think step by step.
Query: {{QUERY}}
KB: {{KB}}
[Examples if few-shot:]
```

**Prompt for Geographical QA Task (Zero-shot and Few-shot CoT)**

```
Task: You will be given a query about geographical entities in
the form of a first order logic predicate that is either True or
False. Your task is to identify whether the answer to the query
is "True" or "False" and also provide a proof of the query. First
state your proof mentioning the rules you used and then say:
"Therefore, the answer is True" or "Therefore, the answer is
False". Think step by step.
Query: {{QUERY}}
[Examples if few-shot:]
```

**Prompt for Geographical QA Task (RAG with Zero-shot and Few-shot CoT)**

```
Task: You will be given a query about geographical entities in
the form of a first order logic predicate that is either True or
False, and a Knowledge Base containing a set of rules that will
help you identify the answer. Your task is to identify whether
the answer to the query is "True" or "False" and also provide a
proof of the query using the knowledge base. First state your
proof mentioning the rules you used and then say: "Therefore, the
answer is True" or "Therefore, the answer is False". Think step
by step.
Query: {{QUERY}}
KB: {{KB}}
[Examples if few-shot:]
```

# H  ANECDOTAL EXAMPLES

To offer more insight into the responses and proofs provided by LLM-TH and the comparison baselines, this section presents anecdotal examples illustrating each model's performance on the evaluated tasks. Since Llama3-70B shows the best overall performance for our comparison baselines in the experimental results provided in Section 5, we show the exemplar outputs from this LLM. Specifically, we showcase the outputs from the following models:

- LLM-TRes
- Zero-shot Chain of Thought prompting
- Few-shot Chain of Thought prompting
- RAG + Zero-shot Chain of Thought prompting
- RAG + Few-shot Chain of Thought prompting

We apply these models to the Deductive reasoning and Geographical QA tasks studied in RQ1 and RQ2 and provide anecdotal examples to elucidate their capabilities and limitations. In each anecdotal example, we present the query, the set of relevant facts and axioms from the KB, and the ground truth proof, followed by the response that each model provides. We also indicate correct proof steps with green color and highlight incorrect ones in red. It is noteworthy that since RQ2 requires access to the KB and studies the model's ability to compensate for KB incompleteness, the only relevant baselines are RAG-based models.

Examples of the experiments on the complete KB are provided in section H.1, and examples of performance on the incomplete KB are detailed in section H.2.

## H.1  EXPERIMENTS ON THE COMPLETE KB

### H.1.1  Deductive Reasoning

---

**Query**: Does Sulfadiazine disinfect Rhodospirillaceae?

---

**Relevant Rules from the KB**:

Facts:

(1) *"Subclass of"*(*"Rhodospirillaceae"* *"Rhodospirillaceae bacteria"*),

(2) *"Gram Negative"*(*"Rhodospirillaceae bacteria"*)

(3) *"Treats Gram Negative"*(*"Sulfadiazine"*)

Axioms:

(4)

$\forall A, B, C,$ *"Subclass of"*$(A, B) \wedge$ *"Gram Negative"*$(B) \wedge$ *"Treats Gram Negative"*$(C)$
$\implies$ *"Unresistant To"*$(A, C)|A:$ *"Bacteria"*$, B:$ *"Taxon"*$, C:$ *"Antibiotic"*

(5)

$\forall A, B,$ *"Unresistant To"*$(A, B) \implies$ *"Can be Used to Treat"*$(A, B)|A:$ *"Bacteria"*,
*"B: Antibiotic"*

(6)

$\forall A, B,$ *"Can be Used to Treat"*$(A, B) \implies$ *"Can be Used to Kill"*$(A, B)|A:$ *"Bacteria"*,
*"B: Antibiotic"*

(7) $\forall A, B,$ *"Can be Used to Kill"*$(A, B) \implies$ *"Disinfects"*$(A, B)|A:$ *"Bacteria"*, *"B: Antibiotic"*

**Ground Truth Proof**:

1- Applying (1), (2), (3) to (4) with the following substitution:

$\theta = \{A/$*"Rhodospirillaceae"*$, B/$*"Rhodospirillaceae bacteria"*$, C/$*"Sulfadiazine"*$\}$ yields:

*"Unresistant To"*(*"Rhodospirillaceae"*, *"Sulfadiazine")*.

2- Applying *"Unresistant To"*(*"Rhodospirillaceae"*, *"Sulfadiazine")* to (5)

with $\theta = \{A/$*"Rhodospirillaceae"*$, B/$*"Sulfadiazine"*$\}$ yields:

*"Can be Used to Treat"*(*"Rhodospirillaceae"*, *"Sulfadiazine")*

3-Applying *"Can be Used to Treat"*(*"Rhodospirillaceae"*, *"Sulfadiazine")* to (6)

with $\theta = \{A/$*"Rhodospirillaceae"*$, B/$*"Sulfadiazine"*$\}$ yields:

*"Can be Used to Kill"*(*"Rhodospirillaceae"*, *"Sulfadiazine")*

4- Applying *"Can be Used to Kill"*(*"Rhodospirillaceae"*, *"Sulfadiazine")* to (7)

with $\theta = \{A/$*"Rhodospirillaceae"*$, B/$*"Sulfadiazine"*$\}$ yields:

5- *"Disinfects"*(*"Rhodospirillaceae"*, *"Sulfadiazine")*.

Therefore, the answer to the query is True.

---

**LLM-TH:**

query: *"Disinfects"*(*"Rhodospirillaceae"*, *"Sulfadiazine"*):

Negated query: $\neg$*"Disinfects"*(*"Rhodospirillaceae","Sulfadiazine")*

Note: In the following sequent calculus formulae, we use *"R"* to refer to *"Rhodospirillaceae"*, *"S"* to represent *"Sulfadiazine"*, *"GN"* to show *"Gram Negative"*, *"TGN"*, to denote *"Treats Gram Negative"*, and *"SO"* to show *"Subclass Of"* due to space limitations.

Step 1-

$$\frac{\neg\text{``Disinfects''}(\text{``R''},\text{``S''}) \quad \neg\text{``Can be Used to Kill''}(A,B) \vee \text{``Disinfects''}(A,B)}{\neg\text{``Can be Used to Kill''}(\text{``R''},\text{``S''})}.$$

Under unification $\theta = \{A/$*"Rhodospirillaceae"*$, B/$*"Sulfadiazine"*$\}$.

Step 2-

$$\frac{\neg\text{``Can be Used to Kill''}(\text{``R''},\text{``S''}) \quad \text{``Can be Used to Kill''}(A,B) \vee \neg\text{``Can be Used to Treat''}(A,B)}{\neg\text{``Can be Used to Treat''}(\text{``R''},\text{``S''})\text{''}}.$$

Under unification $\theta = \{A/$*"Rhodospirillaceae"*$, B/$*"Sulfadiazine"*$\}$.

Step 3-

$$\frac{\neg\text{``Can be Used to Treat''}(\text{``R''},\text{``S''}) \quad \text{``Can be Used to Treat''}(A,B) \vee \neg\text{``Unresistant To''}(A,B)}{\neg\text{``Unresistant To''}(\text{``R''},\text{``S''})}.$$

Step 4-

$$\frac{\neg\text{``Unresistant To''}(\text{``R''},\text{``S''}) \quad \text{``Unresistant To''}(A,B) \vee \neg\text{``SO''}(A,B) \vee \neg\text{``GN''}(B) \vee \neg\text{``TGN''}(C)}{\neg\text{``SO''}(\text{``R''},B) \vee \neg\text{``GN''}(B) \vee \neg\text{``TGN''}(\text{``S''})}$$

Under unification $\theta = \{A/$*"Rhodospirillaceae"*$, C/$*"Sulfadiazine"*$\}$.

Step 5-

$$\frac{\neg ``SO"(``R",B) \lor \neg ``GN"(B) \lor \neg ``TGN"(``S") \quad ``SO"(``R", ``R\ bacteria") \land ``GN"(``R\ bacteria") \land ``TGN"(``S")}{\perp} .$$

Proof score: $\rho_{\text{query}} = (1, 1, 5)$

$$* * *$$

---

**Zero-shot CoT:**

To prove the answer to the query, I will use the following rules from the knowledge base:
1. Rhodospirillaceae is a type of Bacteria. (IsA(Rhodospirillaceae, Bacteria))

2. Sulfadiazine is a type of Antibiotic. (IsA(sulfadiazine, Antibiotic))

3. Antibiotics disinfect Bacteria. (Disinfects(Antibiotic, Bacteria))
Using modus ponens on rules 1, 2, and 3, I can infer that: Disinfects(sulfadiazine, Rhodospirillaceae)
Therefore, the answer is True.

---

**Few-shot CoT:**

1.Sulfadiazine is an antibacterial drug.

2. Rhodospirillaceae is a family of bacteria.

3. Sulfadiazine is not typically used as a disinfectant, but rather as an antibiotic to treat bacterial infections.

Therefore, the answer is False.

---

**RAG + Zero-shot CoT:**

1. SubClassOf(Rhodospirillaceae, Rhodospirillaceae bacteria) (KB)

2. GramNegative(Rhodospirillaceae bacteria) (KB)

3. ∀ A:bacteria, B:Taxon, C:antibiotic, SubClassOf(A, B) && GramNegative(B) && TreatsGram-Negative(C) ⟹ UnResistantTo(A, C) (KB)

4. UnResistantTo(Rhodospirillaceae, sulfadiazine) (inferred from 1, 2, and 3, with A = Rhodospirillaceae, B = Rhodospirillaceae bacteria, and C = sulfadiazine)

5. ∀ A:bacteria B:antibiotic, UnResistantTo(A, B) ⟹ CanBeUsedToTreat(A, B) (KB)

6. CanBeUsedToTreat(Rhodospirillaceae, sulfadiazine) (inferred from 4 and 5)

7. ∀ A:bacteria B:antibiotic, CanBeUsedToTreat(A, B) ⟹ CanBeUsedToKill(A, B) (KB)

8. CanBeUsedToKill(Rhodospirillaceae, sulfadiazine) (inferred from 6 and 7)

9. ∀ A:bacteria B:antibiotic, CanBeUsedToKill(A, B) ⟹ Disinfects(A, B) (KB)

10. Disinfects(Rhodospirillaceae, sulfadiazine) (inferred from 8 and 9)

Therefore, the answer is True.

---

**RAG + Few-shot CoT:**

1. Rhodospirillaceae is a gram negative bacteria,

2. sulfadiazine treats gram negative bacteria,

3. and sulfadiazine can be used to treat gram negative bacteria, therefore sulfadiazine can be used to treat Rhodospirillaceae.

Therefore, the answer is True.

---

**Discussion:**

These anecdotal examples demonstrate that on a complete knowledge base (KB), LLM-TH can successfully prove the query by resolving complementary clauses, without requiring the LLM to generate any additional axioms.

An analysis of the proofs generated by the baseline methods reveals limitations in LLM reasoning, particularly in zero-shot and few-shot settings. For instance, the zero-shot CoT baseline asserts that "Antibiotics disinfect bacteria"; however, this statement is overly broad, as not all antibiotics can disinfect all types of bacteria. While the final answer in this case is correct, there is a flaw in one of the reasoning steps. Also, the few-shot CoT baseline produces an incorrect final answer due to an error in reasoning. In contrast, the RAG-based baselines, utilizing both prompting styles, correctly prove the query by leveraging retrieved facts from the KB.

### H.1.2   Geographical QA

**Query**: Can I drive between Biyabanak and Ryam Frolovskiy?

**Relevant Rules from the KB**:

Facts:

(1) *"Located in"*(*"Biyabanak"*, *"Iran"*),

(2) *"Located in"*(*"Ryam Frolovskiy"*, *"Russia"*),

(3) *"Share Land Borders"*(*"Iran"*, *"Azerbaijan"*)

(4) *"Share Land Borders"*(*"Azerbaijan"*, *"Russia"*)

Axioms:

(5)

$$\forall A, B, \textit{"Share Land Borders"}(A, B) \implies \textit{"Land Connected"}(A, B) | A : country, B : country$$

(6)

$$\forall A, B, C, \textit{"Land Connected"}(A, B) \wedge \textit{"Land Connected"}(B, C)$$
$$\implies \textit{"Land Connected"}(A, C) | A : location, B : location, C : location$$

(7)

$$\forall A, B, C, D, \textit{"Located in"}(A, C) \wedge \textit{"Located in"}(B, D) \wedge \textit{"Land Connected"}(C, D)$$
$$\implies \textit{"Can Drive Between"}(A, B) | A : location, B : location, C : location, D : location$$

**Ground Truth Proof**:

1- Applying (3) to (5) with the following substitution:

$\theta = \{A/\textit{"Iran"}, B/\textit{"Azerbaijan"}\}$ yields:

(8) *"Land Connected"*(*"Iran"*, *"Azerbaijan"*).

2- Applying (3) to (5) with the following substitution:

$\theta = \{A/\textit{"Azerbaijan"}, B/\textit{"Russia"}\}$ yields:

(9) *"Land Connected"*(*"Azerbaijan"*, *"Russia"*).

3- Applying (8), (9) to (6) with the following substitution:

$\theta = \{A/\textit{"Iran"}, B/\textit{"Azerbaijan"}, C/\textit{"Russia"}\}$ yields:

(10) *"Land Connected"*(*"Iran"*, *"Russia"*).

4- Applying (1), (2), (10) to (7) with the following substitution:

$\theta = \{A/\text{``Biyabanak''}, B/\text{``Ryam Frolovskiy''}, C/\text{``Iran''}, D/\text{``Russia''}\}$ yields:

(11) *"Can Drive Between"*("Biyabanak", "Ryam Frolovskiy").

Therefore, the answer to the query is True.

**LLM-TH:**

Proof for query *"Can Drive Between"*("Biyabanak", "Ryam Frolovskiy")

Negated query: $\neg$*"Can Drive Between"*("Biyabanak", "Ryam Frolovskiy")

Note: In the following sequent calculus formulae, we use *"Bi"* to refer to *"Biyabanak"*, *"RF"* to represent *"Ryam Frolovskiy"*, *"Az'"* to refer to '*Azerbaijan*', *"CDB"* to show *"Can Drive Between"*, *"LI"* to show *"Located In"*, *"LC"* to show *"Land Connected"*, and *"SLB"* to show *""Share Land Borders""* due to space limitations.

Step 1-

$$\frac{\neg\text{``CDB''}(\text{``Bi''},\text{``RF''}) \quad \text{``CDB''}(A,B) \vee \neg\text{``LI''}(A,C) \vee \neg\text{``LI''}(B,D) \vee \neg\text{``LC''}(C,D)}{\neg\text{``LI(``Bi''},C)\text{''} \vee \neg\text{``LI''}(\text{``RF''},D) \vee \neg\text{``LC''}(C,D)} \quad .$$

Under unification $\theta = \{A/\text{``Biyabanak''}, C/\text{``Ryam Frolovskiy''}\}$.

Step 2-

$$\frac{\neg\text{``LI''}(\text{``Bi''},C) \vee \neg\text{``LI''}(\text{``RF''},D) \vee \neg\text{``LC''}(C,D) \quad \text{``LI''}(\text{``Bi''},\text{``Iran''}) \wedge \text{``LI''}(\text{``RF''},\text{``Russia''})}{\neg\text{``LC''}(\text{``Iran''},\text{``Russia''})} \quad .$$

Under unification $\theta = \{C/\text{``Iran''}, D/\text{``Russia''}\}$.

Step 3-

$$\frac{\neg\text{``LC''}(\text{``Iran''},\text{``Russia''}) \quad \text{``LC''}(A,C) \vee \neg\text{``LC''}(A,B) \vee \neg\text{``LC''}(B,C)}{\neg\text{``LC''}(\text{``Iran''},B) \vee \neg\text{``LC''}(B,\text{``Russia''})} \quad .$$

Under unification $\theta = \{A/\text{``Iran''}, C/\text{``Russia''}\}$.

Step 4-

$$\frac{\neg\text{``LC''}(\text{``Iran''},B) \quad \text{``LC''}(\text{``Iran''},B) \vee \neg\text{``SLB''}(\text{``Iran''},B)}{\neg\text{``SLB''}(\text{``Iran''},B)} \quad .$$

Step 5-

$$\frac{\neg\text{``LC''}(B,\text{``Russia''}) \quad \text{``LC''}(B,\text{``Russia''}) \vee \neg\text{``SLB''}(B,\text{``Russia''})}{\neg\text{``SLB''}(B,\text{``Russia''})} \quad .$$

Step 6-

$$\frac{\neg\text{``SLB''}(\text{``Iran''}, B) \vee \neg\text{``SLB''}(B, \text{``Russia''}) \quad \text{``SLB''}(\text{``Iran''},\text{``Az''}) \wedge \text{``SLB''}(\text{``Az''},\text{``Russia''})}{\bot} \quad .$$

Under unification $\theta = \{B/\text{``Az''}\}$.

Proof score: $\rho_{\text{query}} = (1, 1, 6)$

$* * *$

**Zero-shot CoT:**

To answer this query, I will use the following rules from the knowledge base:
1. CanDriveBetween(x, y) $\implies$ Connected(x, y)

2. Connected(Biyabanak, Ryam Frolovskiy) $\implies$ False (this is a fact in the knowledge base)

Using rule 1, I can infer that CanDriveBetween(Biyabanak, Ryam Frolovskiy) implies Connected(Biyabanak, Ryam Frolovskiy).
However, according to rule 2, Connected(Biyabanak, Ryam Frolovskiy) is False.
Therefore, the answer is False.

**Few-shot CoT:**

1.Biyabanak is a desert in Iran,

2. and Ryam Frolovskiy is an island in the Caspian Sea, which is not connected to Iran by land.

Therefore, the answer is False.

**RAG + Zero-shot CoT:**

1. LocatedIn(Biyabanak, Iran) (Given in KB)

2. PartOf(Iran, Iran) ( Rule: $\forall$ A:place or location B:place or location, LocatedIn(A, B) $\implies$ PartOf(A, B) )

3. ShareLandBorders(Iran, Azerbaijan) (Given in KB)

4. LandConnected(Iran, Azerbaijan) ( Rule: $\forall$ A:country B:country, ShareLandBorders(A, B) $\implies$ LandConnected(A, B) )

5. LocatedIn(Ryam Frolovskiy, Azerbaijan) (Assuming this is true, since it's not given in KB)

6. PartOf(Ryam Frolovskiy, Azerbaijan) ( Rule: $\forall$ A:place or location B:place or location, LocatedIn(A, B) $\implies$ PartOf(A, B) )

7. CanDriveBetween(Biyabanak, Ryam Frolovskiy) ( Rule: $\forall$ A, B, C, D, LocatedIn(A, C) && LocatedIn(B, D) && LandConnected(C, D) $\implies$ CanDriveBetween(A, B)|A:location, B:location, C:location, D:location )

Therefore, the answer is True.

**RAG + Few-shot CoT:**

1. Since Biyabanak is a desert in Iran,

2. and Ryam Frolovskiy is a place in Russia,

3.and Iran and Russia are land connected (although not explicitly stated in the provided KB, but it can be inferred from the general knowledge that Iran and Russia share a border), then it is possible to drive between Biyabanak and Ryam Frolovskiy.

Using the rule: $\forall$ A, B, C, D , LocatedIn(A, ?C) && LocatedIn(B, ?D) && LandConnected(?C, ?D) $\implies$ CanDriveBetween(A, B) With A = Biyabanak, B = Ryam Frolovskiy, C = Iran, D = Russia

Therefore, the answer is True.

**Discussion:**

Similar to the deductive reasoning experiment, LLM-TH produces the correct proof on a complete knowledge base (KB) without requiring the LLM to surface commonsense axioms from it.
This example also highlights the limitations of the LLM-based baselines. For instance, the zero-shot CoT baseline incorrectly claims that *Biyabanak* and *Ryam Frolovskiy* are not connected, leading to an incorrect final answer. Similarly, the few-shot CoT baseline provides an inaccurate statement about *Ryam Frolovskiy*, describing it as an island in the *Caspian sea* when it is, in fact, a forest in *Russia*.
The RAG-based baselines further reveal challenges in the retrieval stage. For example, the relevant fact about the location of *Ryam Frolovskiy* is not retrieved, prompting the LLM to make assumptions in the zero-shot setting. Although the final answer in this case is correct, the reasoning steps contain errors and deviate from the correct proof. In the few-shot prompting setting, while the final answer is also correct, it is based on an inaccurate statement (in fact, Iran and Russia do not share a land border).

## H.2 EXPERIMENTS ON THE INCOMPLETE KB

### H.2.1 Deductive Reasoning

---

**Query**: Does Sulfadiazine resolve an infection caused by Rhodospirillaceae?

**Relevant Rules from the KB**:

Facts:

(1) *"Subclass of"*(*"Rhodospirillaceae" "Rhodospirillaceae bacteria"*),

(2) *"Gram Negative"*(*"Rhodospirillaceae bacteria"*)

(3) *"Treats Gram Negative"*(*"Sulfadiazine"*)

Axioms:

(4)

$\forall A, B, C,$ *"Subclass of"*$(A, B) \land$ *"Gram Negative"*$(B) \land$ *"Treats Gram Negative"*$(C)$
$\implies$ *"Unresistant To"*$(A, C)|A:$ *"Bacteria"*, $B:$ *"Taxon"*, $C:$ *"Antibiotic"*

(5)

$\forall A, B,$ *"Unresistant To"*$(A, B) \implies$ *"Can be Used to Treat"*$(A, B)|A:$ *"Bacteria"*,
*"B: Antibiotic"*

(6)

$\forall A, B,$ *"Can be Used to Treat"*$(A, B) \implies$ *"Can be Used to Kill"*$(A, B)|A:$ *"Bacteria"*,
*"B: Antibiotic"*

(7) $\forall A, B,$ *"Can be Used to Kill"*$(A, B) \implies$ *"Disinfects"*$(A, B)|A:$ *"Bacteria"*, *"B: Antibiotic"*

---

**Ground Truth Proof**:

1- Applying (1), (2), (3) to (4) with the following substitution:

$\theta = \{A/$*"Rhodospirillaceae"*$, B/$*"Rhodospirillaceae bacteria"*$, C/$*"Sulfadiazine"*$\}$ yields:

*"Unresistant To"*(*"Rhodospirillaceae"*, *"Sulfadiazine"*).

2- Applying *"Unresistant To"*(*"Rhodospirillaceae"*, *"Sulfadiazine"*) to (5)

with $\theta = \{A/$*"Rhodospirillaceae"*$, B/$*"Sulfadiazine"*$\}$ yields:

*"Can be Used to Treat"*(*"Rhodospirillaceae"*, *"Sulfadiazine"*)

3-Applying *"Can be Used to Treat"*(*"Rhodospirillaceae"*, *"Sulfadiazine"*) to (6)

with $\theta = \{A/$*"Rhodospirillaceae"*$, B/$*"Sulfadiazine"*$\}$ yields:

*"Can be Used to Kill"*(*"Rhodospirillaceae"*, *"Sulfadiazine"*)

4- Applying *"Can be Used to Kill"*(*"Rhodospirillaceae"*, *"Sulfadiazine"*) to (7)

with $\theta = \{A/$*"Rhodospirillaceae"*$, B/$*"Sulfadiazine"*$\}$ yields:

5- *"Disinfects"*(*"Rhodospirillaceae"*, *"Sulfadiazine"*).

6- Using the commonsense axiom (absent from the KB) that:

$$\forall A, B, \text{"Disinfects"}(A, B)$$
$$\implies \text{"Resolves Infection Caused By"}(A, B)|A: \text{"Bacteria"}, \text{"B: Antibiotic"}$$

Therefore, the answer to the query is True.

---

**LLM-TH:**

Proof for query *"Resolves Infection Caused By"*(*"Rhodospirillaceae"*, *"Sulfadiazine"*):

Negated query: ¬*"Resolves Infection Caused By"*(*"Rhodospirillaceae"*,*"Sulfadiazine"*)
Note: In the following sequent calculus formulae, we use *"R"* to refer to *"Rhodospirillaceae"*, *"S"* to represent *"Sulfadiazine"*, *"GN"* to show *"Gram Negative"*, *"TGN"*, to denote *"Treats Gram Negative"*, *"SO"* to show *"Subclass Of"* due to space limitations.

Step 1-

$$\frac{\neg \text{"Resolves Infection Caused By"}(\text{"R"},\text{"S"}) \quad \text{"Disinfects(A,B)"} \vee \neg \text{"Can be Used to Kill(A,B)"}}{\neg \text{"Can be Used to Kill"}(\text{"R"},\text{"S"})}.$$

Under unification $\theta = \{A/\text{"Rhodospirillaceae"}, B/\text{"Sulfadiazine"}\}$.

In this step, by theory resolving ¬*"Resolves Infection Caused By"*(*"R"*,*"S"*) with *"Disinfects"(A,B)*, the LLM successfully surfaced the commonsense axiom:

$$\forall A, B, \text{"Disinfects"}(A, B)$$
$$\implies \text{"Resolves Infection Caused By"}(A, B)|A: \text{"Bacteria"}, \text{"B: Antibiotic"}.$$

Step 2-

$$\frac{\neg \text{"Can be Used to Kill"}(\text{"R"},\text{"S"}) \quad \text{"Can be Used to Kill"}(A,B) \vee \neg \text{"Can be Used to Treat"}(A,B)}{\neg \text{"Can be Used to Treat}(\text{"R"},\text{"S"})\text{"}}.$$

Under unification $\theta = \{A/\text{"Rhodospirillaceae"}, B/\text{"Sulfadiazine"}\}$.

Step 3-

$$\frac{\neg \text{"Can be Used to Treat"}(\text{"R"},\text{"S"}) \quad \text{"Can be Used to Treat"}(A,B) \vee \neg \text{"Unresistant To"}(A,B)}{\neg \text{"Unresistant To}(\text{"R"},\text{"S"})\text{"}}.$$

Step 4-

$$\frac{\neg \text{"Unresistant To"}(\text{"R"},\text{"S"}) \quad \text{"Unresistant To"}(A,B) \vee \neg \text{"SO"}(A,B) \vee \neg \text{"GN"}(B) \vee \neg \text{"TGN"}(C)}{\neg \text{"SO"}(\text{"R"},B) \vee \neg \text{"GN"}(B) \vee \neg \text{"TGN"}(\text{"S"})}$$

Under unification $\theta = \{A/\text{"Rhodospirillaceae"}, C/\text{"Sulfadiazine"}\}$.

Step 5-

$$\frac{\neg \text{"SO"}(\text{"R"},B) \vee \neg \text{"GN"}(B) \vee \neg \text{"TGN"}(\text{"S"}) \quad \text{"SO"}(\text{"R"}, \text{"R bacteria"}) \wedge \text{"GN"}(\text{"R bacteria"}) \wedge \text{"TGN"}(\text{"S"})}{\bot}.$$

Therefore, the query is proven to be True with the Proof score: $\rho_{\text{query}} = (1, 0.981, 5)$

$* * *$

---

**RAG + Zero-shot CoT:**

1. SubClassOf(Rhodospirillaceae, Rhodospirillaceae bacteria) (from KB)

2. GramNegative(Rhodospirillaceae bacteria) (from KB)

3. ∀A, B, C, SubClassOf(A, B) ∧ GramNegative(B) ∧ TreatsGramNegative(C) $\implies$ $UnResistantTo(A, C)$ (from KB)

4. SubClassOf(Rhodospirillaceae, Rhodospirillaceae bacteria) $\implies$ PartOf(Rhodospirillaceae, Rhodospirillaceae bacteria) (from KB, using the rule ∀A, B, SubClassOf(A, B) $\implies$ PartOf(A, B)|A: animal, B: Taxon)

5. PartOf(Rhodospirillaceae, Rhodospirillaceae bacteria)∧GramNegative(Rhodospirillaceae bacteria)∧ TreatsGramNegative(sulfadiazine) $\implies$ UnResistantTo(Rhodospirillaceae, sulfadiazine) (from 2, 4, and the assumption that sulfadiazine treats Gram-Negative bacteria)

6. ∀A, B, UnResistantTo(A, B) $\implies$ CanBeUsedToTreat(A, B)|A:bacteria, B:antibiotic (from KB)

7. UnResistantTo(Rhodospirillaceae, sulfadiazine) $\implies$ CanBeUsedToTreat(Rhodospirillaceae, sulfadiazine) (from 5 and 6)

8. ∀A, B, CanBeUsedToTreat(A, B) $\implies$ CanBeUsedToKill(A, B)|A: bacteria, B: antibiotic (from KB)

9. CanBeUsedToTreat(Rhodospirillaceae, sulfadiazine) $\implies$ CanBeUsedToKill(Rhodospirillaceae, sulfadiazine) (from 7 and 8)

10. ∀A, B, CanBeUsedToKill(A, B) $\implies$ Disinfects(A, B)|A: bacteria, B: antibiotic (from KB)

11. CanBeUsedToKill(Rhodospirillaceae, sulfadiazine) $\implies$ Disinfects(Rhodospirillaceae, sulfadiazine) (from 9 and 10) Therefore, the answer is True.

---

**RAG + Few-shot CoT:**

1- Rhodospirillaceae is a gram-negative bacteria,

2- and sulfadiazine treats gram-negative bacterias. Therefore, the answer is True.

---

**Discussion:**

This example demonstrates how LLM-TH can perform commonsense reasoning on an incomplete KB. Since no complementary literal for *"Resolves Infection Caused By"* is found, LLM-TH invokes the LLM to perform a theory resolution step. In this step, it correctly selects *"Disinfects"* as the clause with the highest priority for theory resolution, enabling it to provide the correct proof.
In this example, all baselines are also able to produce the correct proof. However, the RAG+zero-shot CoT baseline exhibits minor errors, such as selecting incorrect KB rules in step 4 and neglecting KB facts in step 5, instead basing its reasoning on an assumption.

---

### H.2.2 Geographical QA

**Query**: Can I take a road trip between Ryam Frolovskiy and Biyabanak?

**Relevant Rules from the KB**:

Facts:

(1) *"Located in"*(*"Biyabanak"*, *"Iran"*),

(2) *"Located in"*(*"Ryam Frolovskiy"*, *"Russia"*),

(3) *"Share Land Borders"*(*"Iran"*, *"Azerbaijan"*)

(4) *"Share Land Borders"*(*"Azerbaijan"*, *"Russia"*)

Axioms:

(5)

$\forall A, B,$ *"Share Land Borders"*$(A, B) \implies$ *"Land Connected"*$(A, B)|A : country, B : country$

(6)

$\forall A, B, C,$ *"Land Connected"*$(A, B) \wedge$ *"Land Connected"*$(B, C)$
$\implies$ *"Land Connected"*$(A, C)|A : location, B : location, C : location$

(7)

$\forall A, B, C, D,$ *"Located in"*$(A, C) \wedge$ *"Located in"*$(B, D) \wedge$ *"Land Connected"*$(C, D)$
$\implies$ *"Can Drive Between"*$(A, B)|A : location, B : location, C : location, D : location$

---

**Ground Truth Proof**:

1- Applying (3) to (5) with the following substitution:

$\theta = \{A/$*"Iran"*$, B/$*"Azerbaijan"*$\}$ yields:

(8) *"Land Connected"*$($*"Iran"*$,$ *"Azerbaijan"*$)$.

2- Applying (3) to (5) with the following substitution:

$\theta = \{A/$*"Azerbaijan"*$, B/$*"Russia"*$\}$ yields:

(9) *"Land Connected"*$($*"Azerbaijan"*$,$ *"Russia"*$)$.

3- Applying (8), (9) to (6) with the following substitution:

$\theta = \{A/$*"Iran"*$, B/$*"Azerbaijan"*$, C/$*"Russia"*$\}$ yields:

(10) *"Land Connected"*$($*"Iran"*$,$*"Russia"*$)$.

4- Applying (1), (2), (10) to (7) with the following substitution:

$\theta = \{A/$*"Biyabanak"*$, B/$*"Ryam Frolovskiy"*$, C/$*"Iran"*$, D/$*"Russia"*$\}$ yields:

(11) *"Can Drive Between"*$($*"Biyabanak"*$,$ *"Ryam Frolovskiy"*$)$.

5- Using the commonsense axiom (absent from the KB) that:

$\forall A, B,$ *"Can Drive Between"*$(A, B)$
$\implies$ *"Can Take Road Trip Between"*$(A, B)|A:$ *"location"*$, $*"B: location"*

Therefore, the answer to the query is True.

---

**LLM-TH:**

Proof for query *"Can Drive Between"*$($*"Biyabanak"*$,$ *"Ryam Frolovskiy"*$)$

Negated query: $\neg$*"Can Drive Between"*$($*"Biyabanak"*$,$ *"Ryam Frolovskiy"*$)$

Note: In the following sequent calculus formulae, we use *"Bi"* to refer to *"Biyabanak"*, *"RF"* to represent *"Ryam Frolovskiy"*, *"Az'"* to refer to *'Azerbaijan"*, *"CTRTP"* to show *"Can Take Road Trip Between"*, *"CDB"* to show *"Can Drive Between"*, *"LI"* to show *"Located In"*, *"LC"* to show *"Land Connected"*, and *"SLB"* to show *""Share Land Borders""* due to space limitations.

Step 1-

$$\frac{\neg\text{*"CTRTB"*}(\text{*"Bi"*},\text{*"RF"*}) \quad \text{*"CDB"*}(A,B) \vee \neg\text{*"LI"*}(A,C) \vee \neg\text{*"LI"*}(B,D) \vee \neg\text{*"LC"*}(C,D)}{\neg\text{*"LI("Bi",C)"*} \vee \neg\text{*"LI"*}(\text{*"RF"*},D) \vee \neg\text{*"LC"*}(C,D)}$$ .

Under unification $\theta = \{A/$*"Biyabanak"*$, C/$*"Ryam Frolovskiy"*$\}$.

In this step, by theory resolving ¬*"Can Take Road Trip Between"("Bi", "RF")* with *"Can Drive Between"(A,B)*, the LLM successfully surfaced the commonsense axiom:

$\forall A, B,$ *"Can Drive Between"*$(A, B)$

$\implies$ *"Can Take Road Trip Between"*$(A, B)|A:$ *"location"*, *"B: location"*.

Step 2-

$$\frac{\neg\text{``LI''("Bi",C)} \vee \neg\text{``LI''("RF",D)} \vee \neg\text{``LC''(C,D)} \quad \text{``LI''("Bi","Iran")} \wedge \text{``LI''("RF","Russia")}}{\neg\text{``LC''("Iran","Russia")}} .$$

Under unification $\theta = \{C/\text{``Iran''}, D/\text{``Russia''}\}$.

Step 3-

$$\frac{\neg\text{``LC''("Iran","Russia")} \quad \text{``LC''(A,C)} \vee \neg\text{``LC''(A,B)} \vee \neg\text{``LC''(B,C)}}{\neg\text{``LC''("Iran",B)} \vee \neg\text{``LC''(B,"Russia")}} .$$

Under unification $\theta = \{A/\text{``Iran''}, C/\text{``Russia''}\}$.

Step 4-

$$\frac{\neg\text{``LC''("Iran",B)} \quad \text{``LC''("Iran",B)} \vee \neg\text{``SLB''("Iran",B)}}{\neg\text{``SLB''("Iran",B)}} .$$

Step 5-

$$\frac{\neg\text{``LC''(B,"Russia")} \quad \text{``LC''(B,"Russia")} \vee \neg\text{``SLB''(B,"Russia")}}{\neg\text{``SLB''(B,"Russia")}} .$$

Step 6-

$$\frac{\neg\text{``SLB''("Iran", B)} \vee \neg\text{``SLB''(B, "Russia")} \quad \text{``SLB''("Iran","Az")} \wedge \text{``SLB''("Az","Russia")}}{\perp} .$$

Under unification $\theta = \{B/\text{``Az''}\}$.

Therefore, the query is proven to be True with the proof score: $\rho_{\text{query}} = (1, 0.994, 6)$

$* * *$

---

**RAG + Zero-shot CoT:**

1- From the knowledge base, we have: *PartOf(Biyabanak, Iran)* (from the given KB)

2- PartOf(Ryam Frolovskiy, Belarus) (not mentioned in the KB, but assuming it's true for the sake of the query)

3- Using the rule:

"$\forall A, B, C, PartOf(A, B) \wedge PartOf(B, C) \implies PartOf(A, C)|A:$ *location*, $B:$ *location*, $C:$ *location*", We can infer: *PartOf(Biyabanak, Asia)* (since Iran is part of Asia)

4- and *PartOf(Ryam Frolovskiy, Europe)* (since Belarus is part of Europe)

5- Using the rule: $\forall A, B, C, LandConnected(A, B) \wedge LandConnected(B, C) \implies LandConnected(A, C)|A:country, B:country, C:country$, we can infer: *Not LandConnected(Iran, Belarus)* (since Iran is in Asia and Belarus is in Europe, and they are not land-connected)

6- Using the rule:"$\forall A, B, C, D, LocatedIn(A, C) \wedge LocatedIn(B, D) \wedge LandConnected(C, D) \implies CanDriveBetween(A, B)|A: location, B: location, C: country, D: country$", we cannot infer *CanDriveBetween(Biyabanak, Ryam Frolovskiy)* since Iran and Belarus are not land-connected. Therefore, the answer is False.

---

**RAG + Few-shot CoT:**

1- To determine if you can take a road trip between Biyabanak and Ryam Frolovskiy, we need to check if Biyabanak and Ryam Frolovskiy are located in countries that are land connected.
2- From the KB, we know that Biyabanak is part of Iran,

3- and Ryam Frolovskiy is not mentioned in the KB, but since it is not mentioned, we will assume it is in Russia (based on the name).

4- From the KB, we also know that Iran and Russia are land connected (rule 1).

5- Using the rule "$\forall A, B, C, D, LocatedIn(A, C) \wedge LocatedIn(B, D) \wedge LandConnected(C, D) \implies CanDriveBetween(A, B)|A: location, B: location, C: country, D: country$", we can infer that you can take a road trip between Biyabanak and Ryam Frolovskiy. Therefore, the answer is True.

---

**Discussion:**

This example further illustrates the successful application of LLM-based theory resolution by LLM-TH, where a theory resolution step leverages the LLM's commonsense knowledge to address KB incompleteness. Specifically, in Step 1, the LLM surfaces a commonsense axiom to theory resolve ¬*Can Drive Between"* with ¬*Can Take Road Trip Between"*.

This example also highlights the limitations of the RAG-based baselines. Due to a failure in retrieving facts about *Ryam Frolovskiy*, the RAG+Zero-shot CoT baseline makes an incorrect assumption about its location. Furthermore, subsequent reasoning steps are flawed, resulting in a wrong answer. Although the RAG+Few-shot CoT baseline provides the correct final answer, its reasoning process is not entirely reliable. Notably, because the LLM lacks access to a fact about the location of *Ryam Frolovskiy*, it makes an assumption about its location—which happens to be correct in this case. However, it also incorrectly references a rule in the KB that states *Iran* and *Russia* are land connected, even though no such rule exists in the KB.

These examples underscore the limited reliability of existing LLM-based baselines when combining commonsense reasoning with factual information.

