# OpenReview forum: "LLM-based Typed Hyperresolution for Commonsense Reasoning with Knowledge Bases"
_ICLR.cc/2025/Conference — ICLR 2025 Poster_

### Official Review · Reviewer_97vV · 2024-11-01

**Soundness:** 3
**Presentation:** 3
**Contribution:** 2
**Rating:** 6
**Confidence:** 4

**Summary:**

The paper introduces LLM-based Typed Hyperresolution (LLM-TH), a novel framework designed to enhance logical commonsense reasoning in large language models (LLMs). LLM-TH addresses key limitations of existing methods by integrating the internal commonsense knowledge of LLMs with axiomatic knowledge bases, providing a mechanism to repair erroneous inference steps, and enabling reasoning over large knowledge bases with tens of thousands of rules. The framework is built on the concept of "theory resolution," which allows the incorporation of specialized theorem provers into the resolution inference rule, and utilizes *hyperresolution* to efficiently combine clauses for scalable reasoning.

Experiments on Preference reasoning, Multi-domain Deductive reasoning, and Geographical QA, which all requires commonsense reasoning over rules/KBs, prove the effectivenss of the method, outperforming standard RAG/CoT.

**Strengths:**

1. The main contribution is using the entailment ability of language models as substitute for semantic parsing to identifies the unsatisfiable natural language predicates to perform reasoning via theory resolution. The motivation is sound and natural.

2. Sound writing and problem formulation.

**Weaknesses:**

- Simple commonsense reasoning is supposed to be considered as solved by scaling language models. For example, GPT-4o or its predecessors all know the commonsense/entailment knowledge in the examples you provided, “Souvlaki”(y) =⇒ “Meditteranean”(y) and "cuttlefish is not fish" with simple prompts, which kind of make the FIXING ERRONEOUS RESOLUTIONS module useless.

E.g.,
> Prompt: Is this true? ∀y“Souvlaki”(y) =⇒ “Meditteranean”(y)
> Response: The statement ∀y (“Souvlaki”(y) =⇒ “Mediterranean”(y)) can be interpreted as "For all y, if y is Souvlaki, then y is Mediterranean." In general, this statement is true because Souvlaki is a popular Greek dish, and Greece is part of the Mediterranean region. Therefore, if something is identified as Souvlaki, it is reasonable to classify it as Mediterranean cuisine.

- Second, since commonsense is quite simple for LLMs, the major part of performing commonsense reasoning based on axioms is the efficient and effective retrieval of constraints/rules in the KB, while this part is supposed to be well-studied before. Moreover, I am interested in the performance of transforming all clauses, axioms, queries into natural language and ask LLMs to solve the task based on it's own parametric knowledge. I don't think the performance would be significantly lower.

- Missing discussions on commonsense reasoning. Most of "commonsense knowledge" are simple and do not requires complex reasoning, and state-of-the-art LLMs can already capture them after instruction tuning on manually curated commonsense resources. What your work is more closely related to is complex commonsense reasoning, which requires multiple reasoning steps and possibly grounded in logical forms. Checkout works like WinoLogic, COM2 (complex commonsense reasoning over KB), CRoW (commonsense in real world).

**Questions:**

See weaknesses.

---

### Official Review · Reviewer_8yhZ · 2024-11-04

**Soundness:** 2
**Presentation:** 1
**Contribution:** 2
**Rating:** 1
**Confidence:** 4

**Summary:**

This paper introduces LLM-based Typed Hyperresolution (LLM-TH), a novel framework for commonsense reasoning that combines large language models with traditional logical inference methods. The problem that the authors try to address is that current LLMs have limitations in commonsense reasoning tasks due to hallucinations and the inability to handle large knowledge bases. LLM-TH uses theory resolution to integrate LLMs into logical reasoning and implements a typing mechanism to reduce errors and improve efficiency. Meanwhile, LLM-TH also employs hyper-resolution to handle large knowledge bases and a mechanism to repair incorrect inference steps. In experiments, the authors used BART with 406M parameter model, LLM-TH outperformed larger models, such as Llama3-70B, Gemini1.5-Flash, GPT-3.5-Turbo, and MIstral 46.7B on three reasoning tasks: preference reasoning, multi-domain deductive reasoning, and geographical questions answering. However, the writing of this paper doesn't follow a good structure and the readers are hard to follow what are the authors trying to do in each section.

**Strengths:**

1. This paper combines the two approaches to traditional logical reasoning: traditional logical reasoning and LLM's ability to understand commonsense, which is quite novel.
2. The results on three benchmarks show that the model can solve the problem very effectively.
3. The method also provides a theoretical framework for error repair.

**Weaknesses:**

1. The baseline compared in this paper is quite weak, with only some large LLMs like GPT-3.5 and Llama3-70B. You should add more commonsense models, like COMET and Vera.
2. The writing of the paper needs significant improvement. Currently, it is quite hard to follow the paper and understand its content. The authors write the paper with very dense descriptions without clearly a clear logical flow. For example, the authors don't mention each component of the method and their purpose, making the whole method part quite hard to understand, while there are too many equations in the LLM-TH model on page 4 and page 5, which is too dense. (too many details, no brief introduction).
3. The paper needs more case studies about why the LLM-TH can perform much better than solely LLMs. The authors need more motivations to justify why we need to use those symbolic methods to solve commonsense reasoning, which is not symbolic at all.

**The experiments in this paper are very weak, with only two experiments on complete and incomplete graphs. There is literally no analysis of why their methods can work so well, given the authors introduced so many components in their framework (the repair mechanism, hyperresolution, and type inference). No ablation study, no case study or error analysis**

**Questions:**

No

---

### Official Review · Reviewer_hYKL · 2024-11-07

**Soundness:** 2
**Presentation:** 2
**Contribution:** 3
**Rating:** 8
**Confidence:** 3

**Summary:**

The paper introduces an LM-based hyper-resolution system LLM-TH that utilizes the language models' common sense to:
1. augment incomplete KBs: The LM scores entail between literals and add rules that can help to complete the proofs.
2. deduce type matching for typed resolution: The LM scores the consistency between two types to allow unification between literals with different types.

The LM entailment and type-match scores are used to implement a priority queue that allows proofs to be completed faster by finding clauses with a higher chance of being resolved. Moreover, the results require only a small LM to perform the NLI task. The paper uses a BART-large model trained on MNLI.

The method is evaluated on 3 knowledge-based reasoning tasks: Preference Reasoning (over recipes and user requests), Deductive Reasoning (a new task over five domains), and Geographical QA (a new task of geographical reasoning over a KB of geographical facts). The method shows strong performance (compared to larger LLM baselines) on all 3 tasks. Manual inspection shows that the method finds correct proofs, i.e., it is right for the right reasons.

The ability to perform inference with incomplete KBs is shown by removing facts from the KB that are required in the final proof. The LLM-TH system is able to recover (near perfectly) from the incomplete KB.

---

**Update after rebuttal:** The authors sufficiently address my concerns and questions. The new version of the paper clarifies several points of confusion. I am increasing my score from 5 to 8. I have not thoroughly reviewed all the new edits to the paper, just those concerning my questions.

**Strengths:**

- The method is verifiable and faithful by definition. The execution of proofs can be traced back deterministically
- Faulty system reasoning can be modified by inserting corrected (repair) axioms into the knowledge base
    - This ability is demonstrated theoretically (caveat in the later review section) via an example but not tested at scale
- The method scales to large KBs since it does not require the entire KB to fit in the LM context
- Experiments demonstrate that the method can handle incomplete KBs to some extent by compensating for missing facts with LM entailment inference

**Weaknesses:**

1. The reasoning for why any symbolic baselines cannot be implemented is not convincing
    - One reason provided is that past methods require complete knowledge bases. Is that not the setting on Table 1?
2. The complete algorithm of the method is not correctly presented
    - Clarification questions about Algo 1 are in the next section
3. The lack of the complete algorithm and the lack of any timing analysis makes it difficult to judge the feasibility of the system
    - How many LM inferences are required in each resolution step? Theoretical or empirical measurement is necessary
    - I do not believe that high latency (up to some reasonable degree) is grounds for rejecting the paper. However, this paper needs this analysis for completeness
4. It is unclear how the repair axioms for LM entailment scoring are used in reasoning

**Questions:**

1. Algorithm 1: I feel there are several typos and unclear variable roles here
    - What is the point of max_iters? The inner loop completes one entire proof search.
    - The counter i never changes
    - Line 9: Why is max_proofs increasing?
    - Line 10: What is the point of saving the empty clause to proofs?
    - Line 12: Is it invalid to resolve $c$ to a clause with less arity than $c$?
    - Line 13: Is the loop over $\beta_c$?
2. How many LM inferences are performed in each resolution iteration is unclear. Please provide an explicit algorithm for implementing LM-based resolution
3. Proof of Proposition 1: It is not obvious how incorrect LLM belief (invalid linkage of literals based on entailment) can be corrected with an axiom. This confusion again stems from the unclear implementation of the LM hyper-resolution. For the given example, if the LM incorrectly infers that $\text{cuttlefish}(x) \implies \text{fish}(x)$, then how does adding $\text{cuttlefish}(x) \nRightarrow \text{fish}(x)$ correct the reasoning? Is the predicted LM inference checked against all KB facts at every iteration?
4. Please provide examples of query types in the DEductive Reasoning and Geographical QA datasets
5. Sec 5.2.2: When reasoning with incomplete KBs, does LM-TH add back the removed edge?
6. Sec 5.2.2: When creating the setting with incomplete KBs, does removing a single edge remove all valid proofs? Asked another way, is the KB truly incomplete or it just the "gold" proof invalidated?
7. Sec 5.2.3: The dataset description states that some proofs in the Deductive Reasoning dataset require up to 7 steps. How are the average search steps in Fig 3 about 3.5 with typed resolution? Can you provide a breakdown of this comparison as a function of a number of steps in the "gold" proof?

---

### Official Review · Reviewer_ngRa · 2024-11-08

**Soundness:** 3
**Presentation:** 3
**Contribution:** 3
**Rating:** 6
**Confidence:** 2

**Summary:**

The paper introduces LLM-based Typed Hyperresolution (LLM-TH), a novel framework for enhancing commonsense reasoning by LLMs through logical inference with large, potentially incomplete KBs. The key ideas involve combining theory resolution, where the LLM fills in gaps in the KB by identifying commonsense entailments, with typed hyperresolution, which improves efficiency by limiting reasoning steps to type-consistent paths. This approach addresses the limitations of traditional LLM reasoning methods, which struggle with errors, hallucinations, and scalability to large KBs.

The main contributions are summarised at the end of the introduction section.

**Strengths:**

The paper successfully integrates LLMs with classical logical reasoning methods, leveraging the commonsense knowledge of LLMs to enhance reasoning over incomplete KBs.

The introduction of typed hyperresolution significantly improves the scalability of the reasoning process, making it feasible to handle large-scale KBs.

The framework provides transparency in the reasoning process and offers a reliable method to fix errors, which is crucial for high-stakes applications.

The paper presents a thorough empirical evaluation across multiple tasks and datasets, demonstrating the effectiveness of LLM-TH compared to existing baselines.

**Weaknesses:**

While the framework enhances reasoning accuracy, it remains heavily dependent on the LLM’s commonsense knowledge for entailment inference. This reliance could present challenges if the LLM lacks domain-specific knowledge or displays biases.

Although typing improves search efficiency, ensuring type consistency across large datasets may introduce notable computational overhead, particularly in knowledge bases with complex hierarchical type structures.

**Questions:**

How does the performance of LLM-TH change when paired with different LLMs that may vary in their levels of commonsense knowledge or domain-specific expertise?

How does LLM-TH handle cases where type assignments within the KB are ambiguous or inconsistent?

---

### Meta-Review · Area_Chair_YrLD · 2024-12-21

**Metareview:**

(a) Scientific Claims and Findings
The paper proposes LLM-TH, a novel integration of LLMs with symbolic logic to enhance commonsense reasoning. It employs typed hyperresolution to improve scalability and error correction mechanisms for verifiable reasoning. The framework is evaluated on three reasoning tasks, demonstrating superior accuracy and efficiency compared to large LLM baselines.

(b) Strengths

- Introduces a novel method combining symbolic reasoning and LLMs for scalable and verifiable reasoning.
- Effectively reduces errors and hallucinations, offering theoretical guarantees for error correction.
- Demonstrates strong empirical results on diverse tasks with thorough experiments.
- Efficiently handles large knowledge bases beyond the context size limitations of LLMs.

(c) Weaknesses

- Reliance on the LLM’s commonsense knowledge could limit performance in domain-specific contexts.
- Baseline comparisons with symbolic methods are limited, raising questions about the specific contributions of proposed components (e.g., repair mechanism, type inference).
- Missing ablation studies and detailed error analyses to isolate contributions of individual components.

(d) Decision: accept
The paper addresses a significant limitation in commonsense reasoning by combining LLMs with symbolic inference techniques, presenting strong empirical results. The novelty and potential impact warrant acceptance.

**Additional Comments On Reviewer Discussion:**

Key Points Raised by Reviewers:

- Reviewer ngRa: Commended the method's integration of LLMs with symbolic logic and empirical results but noted concerns about dependency on LLM commonsense and computational overhead.
- Reviewer hYKL: Highlighted strengths in verifiability and scalability but raised questions about algorithmic clarity and missing timing analyses. The rebuttal sufficiently addressed concerns, leading to an increased score.
- Reviewer 8yhZ: Criticized weak baselines, lack of ablation studies, and suspiciously high performance metrics. Maintained a strong reject stance despite detailed rebuttals and additional experiments.
- Reviewer 97vV: Questioned the necessity of the framework given the strength of existing LLMs but acknowledged the soundness of the method. Maintained a marginally positive score.

Author Rebuttal: Authors addressed reviewer concerns with additional experiments, comparisons to new baselines (VERA, Logic-LM, and LINC), and clarifications on algorithmic details.

The reviewers' concerns about experimental design and paper clarity were considered, but the novelty and empirical results ultimately outweighed these limitations. The disagreement between reviewers, particularly Reviewer 8yhZ’s strong rejection, and others' moderate to strong support highlights the divisive nature of the paper.

---

### Decision · Program_Chairs · 2025-01-22

Accept (Poster)